# Faster Differentially Private Convex Optimization via Second-Order Methods

**Arun Ganesh**
Google Research

**Mahdi Haghifam**[*]
University of Toronto,
Vector Institute

**Thomas Steinke**
Google DeepMind

**Abhradeep Thakurta**
Google DeepMind

## Abstract

Differentially private (stochastic) gradient descent is the workhorse of DP private machine learning in both the convex and non-convex settings. Without privacy constraints, second-order methods, like Newton's method, converge faster than first-order methods like gradient descent. In this work, we investigate the prospect of using the second-order information from the loss function to accelerate DP convex optimization. We first develop a private variant of the regularized cubic Newton method of Nesterov and Polyak [NP06], and show that for the class of strongly convex loss functions, our algorithm has quadratic convergence and achieves the optimal excess loss. We then design a practical second-order DP algorithm for the unconstrained logistic regression problem. We theoretically and empirically study the performance of our algorithm. Empirical results show our algorithm consistently achieves the best excess loss compared to other baselines and is 10-40× faster than DP-GD/DP-SGD for challenging datasets.

## 1 Introduction

Many machine learning tasks reduce to a convex optimization problem. More precisely, given a dataset $S_n = (z_1, \ldots, z_n) \in \mathcal{Z}^n$, a closed, convex set $\mathcal{W} \subseteq \mathbb{R}^d$, and a loss function $f : \mathcal{W} \times \mathcal{Z} \to \mathbb{R}$ such that, for every $z \in \mathcal{Z}$, $f(w, z)$ is a convex function in $w$, our goal is to compute an approximation to $\arg\min_{w \in \mathcal{W}} \left( \ell(w, S_n) \triangleq \frac{1}{n} \sum_{i \in [n]} f(w, z_i) \right)$. In this paper, we are interested in the problem of designing optimization algorithms in the scenario that the dataset $S_n$ contains private information. Differential privacy (DP) [DMNS06] is a formal standard for privacy-preserving data analysis that provides a framework for ensuring that the output of an analysis on the data does not leak this private information. This problem is known as *private convex optimization*: Design an algorithm $\mathcal{A} : \mathcal{Z}^n \to \mathcal{W}$ that is both DP and ensures low *excess loss* $\triangleq \ell(\mathcal{A}(S_n), S_n) - \min_{w \in \mathcal{W}} \ell(w, S_n)$.

The predominant algorithm for private convex optimization is DP (stochastic) gradient descent (DP-GD/DP-SGD). This is a *first-order* iterative method. I.e., we start with an initial value $w_0$ and iteratively update it using the gradient of the loss $\nabla_{w_t} \ell(w_t, S_n)$ following the update rule $w_{t+1} = w_t - \eta \cdot (\nabla_{w_t} \ell(w_t, S_n) + \xi_t)$, where $\eta > 0$ is a constant and $\xi_t$ is Gaussian noise to ensure privacy. The number of iterations $T$ also determines the amount of noise at each iteration, i.e., the scale of $\xi_t$ is proportional to $\sqrt{T}$ due to the composition of DP. Note that we assume $\|\nabla_{w_t} \ell(w_t, S_n)\| \leq 1$.

One of the major drawbacks of DP-(S)GD is *slow convergence*. The choice of $(\eta, T)$ exhibits a tradeoff in terms of the excess loss: if $\eta \cdot T$ is small, the algorithm cannot reach the optimal solution; on the other hand, the magnitude of noise at each iteration is $\eta \cdot \sqrt{T}$, which cannot be too large. Therefore, to maximize $\eta \cdot T$ and minimize $\eta \cdot \sqrt{T}$, implementations of DP-(S)GD err on the side of

---

[*]This work was carried out while the author was an intern at Google Research, Brain Team.
m.haghifam@northeastern.edu, {arunganesh, steinke, athakurta}@google.com

large $T$ and small $\eta$, which results in a long, slow path to convergence. This fact has been shown theoretically as well: for the class of $\beta$-smooth convex functions, the optimal instantiations of DP-GD use a step size of $\max\{1/\sqrt{n}, \sqrt{d}/\varepsilon n\}$ [BFTG19] while in the non-private setting the stepsize for GD is set to $1/\beta$. Smaller step size requires more steps (i.e. more iterations) to converge. This slowness is exacerbated by the facts that (1) DP-SGD requires large batch sizes for good performance [PHKX+23] and (2) the hyperparameter tuning of DP-(S)GD, and generally DP algorithms, is a challenging task [PS22]. *Can we design a DP optimization algorithm which accelerates DP-(S)GD by choosing the step size dynamically based on the local geometry of the loss function?*

We draw inspiration from the non-private optimization literature: To address the slow convergence of GD and of first-order methods in general, a class of algorithms based on *preconditioning* the gradient using second-order information has been developed [Nes98; NW99]. This class of algorithms is based on successively minimizing a quadratic *approximation* of the function, i.e., $w_{t+1} = w_t + \Delta_t$ where $\Delta_t = \arg\min_\Delta \{\ell(w_t, S_n) + \langle \nabla\ell(w_t, S_n), \Delta \rangle + \frac{1}{2}\langle H_t \cdot \Delta, \Delta \rangle\} = -(H_t)^{-1}\nabla\ell(w_t, S_n)$. Here, $H_t$ is a scaling matrix which provides curvature information about the loss $\ell(\cdot, S_n)$ at $w_t$. For instance, Newton's method uses the Hessian $H_t = \nabla^2\ell(w_t, S_n)$. Second-order algorithms significantly improve over the convergence speed of GD, and key to their success is that at each step they *automatically* tune the stepsize along each dimension based on the local curvature.

In this paper, our goal is to accelerate DP convex optimization. In particular, the current paper revolves around the following questions: Can the second-order information *accelerate* private convex optimization while achieving *optimal excess error*? What is the best way to *privatize second-order information*, e.g., the Hessian matrix? How does the achievable *privacy-utility-runtime tradeoff* compare with first-order methods such as DP-GD? We show that second-order information can accelerate DP optimization while achieving excess loss that matches or improves on DP-GD. Our main contributions are both theoretical and empirical:

## 1.1 Provably Optimal Algorithm for Strongly Convex Functions

Newton's method is a second-order optimization technique that is well-known for its rapid convergence for strongly convex and smooth functions in non-private optimization. Specifically, to achieve an excess loss of $\alpha$, the method only requires $O(\log\log(1/\alpha))$ iterations, which is provably faster than the convergence rate of *any* first-order method. One natural question is whether it is possible to design a second-order DP convex optimization algorithm that can achieve the *optimal minmax* excess error $\mathrm{err}^{\mathrm{opt}}$ in $O(\log\log(1/\mathrm{err}^{\mathrm{opt}}))$ iterations? We provide an affirmative answer to this question in Section 4 by designing a second-order DP algorithm based on the cubic regularized Newton's method of Nesterov and Polyak [NP06]. At each step $t$, we compute a *cubic* upper bound $\ell(w+\Delta, S_n) \leq \ell(w, S_n) + \langle \nabla_w\ell(w, S_n), \Delta \rangle + \frac{1}{2}\langle \nabla_w^2\ell(w, S_n)\cdot\Delta, \Delta \rangle + O(\|\Delta\|^3)$. We can minimize this cubic upper bound using *any* DP convex optimization subroutine; the minimizer becomes the next iterate $w_{t+1}$. Since the cubic is a universal upper bound, our algorithm converges globally

## 1.2 Fast Practical Algorithms for DP Logistic Regression

DP logistic regression is a popular approach for private classification, with DP-GD/DP-SGD being the predominant class of algorithms for this task. As we numerically show, DP-GD/DP-SGD exhibit slow convergence for this task (See Figure 1). In Section 5, we develop a practical algorithm that injects carefully designed noise into Newton's update rule as follows:

$$w_{t+1} = w_t - \Psi\left(\nabla_{w_t}^2\ell(w_t, S_n)\right)^{-1}\cdot(\nabla_{w_t}\ell(w_t, S_n) + \xi_{t,1}) + \xi_{t,2}. \tag{1}$$

In particular, we inject noise twice: $\xi_{t,1}$ privatizes the gradient and $\xi_{t,2}$ privatizes the direction. The function $\Psi$ modifies the Hessian to ensure that the eigenvalues are not too small; this is essential for bounding the sensitivity and, hence, the scale of $\xi_{t,2}$. We consider two types of modification based on *eigenvalue clipping* and *eigenvalue adding*. For eigenvalue clipping, $\Psi(\nabla_{w_t}^2\ell(w_t, S_n))$ replaces the eigenvalues $\lambda_i$ of $\nabla_{w_t}^2\ell(w_t, S_n)$ with $\max\{\lambda_i, \lambda_0\}$, where $\lambda_0 > 0$ is a carefully chosen constant. For eigenvalue adding, $\Psi(\nabla_{w_t}^2\ell(w_t, S_n)) = \nabla_{w_t}^2\ell(w_t, S_n) + \lambda_0 I$. Using $\Psi$ we can control the sensitivity and still have fast convergence, since important curvature information is generally

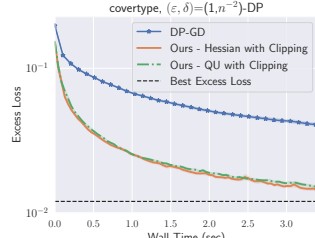

Figure 1: Excess loss versus runtime of DP-GD & our algorithms.

contained in the larger eigenvalues/vectors of the Hessian. We prove the local convergence of the update rule (1) in Section 5.3 and perform a thorough empirical evaluation Section 6. We demonstrate that our algorithm outperforms existing baselines on a variety of benchmarks.

**Ensuring Global Convergence.** One limitation of the update rule in Equation (1) is it does not converge globally (even without noise added for DP). That is, if the initial point $w_0$ is too far from the optimal solution, then the iterates may diverge. To address this problem, we propose a variant of Newton's update rule where we replace the Hessian with a different form of second-order information which gives a *Quadratic Upperbound* (QU) on the logistic loss. This is *guaranteed to converge globally*, like the cubic Newton approach. And we show numerically that this algorithm converges almost as fast as the regular Newton's method in the private setting. Figure 1 shows the convergence speed of our algorithms and DP-GD in terms of real wall time for the task of logistic regression on the Covertype dataset for $(\varepsilon, \delta) = (1, (\text{num. samples})^{-2})$-DP. Despite DP-GD having a lower per-iteration cost, our algorithm is $30\times$ faster than DP-GD and achieves better excess loss.

**Stochastic Minibatch Variant.** We also show that our algorithms naturally extend to the minibatch setting where gradient and second-order information are computed on a subset of samples. We numerically compare it with DP-SGD and show that it has faster convergence.

## 2 Related Work

DP optimization is a well-studied topic [e.g., SCS13; MRTZ17; ACGM+16; STU17; WLKC+17; INST+19; STT20; SSTT21; GTU22; GLL22; BFTG19; BST14]. Most similar to our work, Avella-Medina, Bradshaw, and Loh [ABL21] consider second-order methods for DP convex optimization. We provide a detailed comparison between our results and theirs in Remark 4.5 and Section 6 showing that our algorithms relax restrictive assumptions and provide better excess error for logistic regression.

There are numerous non-private second-order optimization methods in the literature. The choice of method depends primarily on the values of $n$ and $d$. When $n$ is large, several works consider various sampling techniques for constructing second-order information, see [RM19; XYRRM16; Erd15; EM15]. When $d$ is large, various methods are proposed in the literature for efficient approximation of the Hessian matrix, see [ABH17; Erd15; EM15; XYRRM16; GKLR19]. There is also a family of algorithms based on the estimation of the curvature from the change in gradients. These algorithms are generally known as quasi-Newton methods stemming from the seminal BFGS algorithm [JM23].

## 3 Preliminaries

Let $d \in \mathbb{N}$. For a vector $x \in \mathbb{R}^d$, $\|x\|$ denotes the $\ell_2$ norm of $x$. Let $n, m \in \mathbb{N}$. For a matrix $A \in \mathbb{R}^{n \times m}$, $\|A\| = \sup_{x \in \mathbb{R}^m : \|x\| \leq 1} \|Ax\|$ denotes the operator norm, and $\|A\|_F \triangleq \sqrt{\text{trace}(A^T \cdot A)}$ denotes the Frobenius norm of $A$ where trace denotes the trace operator. $I_d \in \mathbb{R}^{d \times d}$ denotes the identity matrix. $\langle \cdot, \cdot \rangle$ denotes the standard inner product in $\mathbb{R}^d$. For a convex and closed subset $\mathcal{W} \subseteq \mathbb{R}^d$, let $\Pi_{\mathcal{W}} : \mathbb{R}^d \to \mathcal{W}$ be the Euclidean projection operator, given by $\Pi_{\mathcal{W}}(x) = \arg\min_{y \in \mathcal{W}} \|y - x\|_2$. For a (measurable) space $\mathcal{R}$, $\mathcal{M}_1(\mathcal{R})$ denotes the set of all probability measures on $\mathcal{R}$. Note that the statements in the paper about random variables hold almost surely. We will skip such declarations to aid readability. Let $\mathcal{Z}$ be the data and let $\mathcal{W} \subseteq \mathbb{R}^d$ be the parameter space. Let $f : \mathcal{W} \times \mathcal{Z} \to \mathbb{R}$ be a loss function. Throughout the paper, we assume $f$ is doubly continuous, a convex function in $w$, and $\mathcal{W}$ is a closed and convex set. We say (1) $f$ is $\mathrm{L}_0$-*Lipschitz* iff there exists $\mathrm{L}_0 \in \mathbb{R}$ such that $\forall z \in \mathcal{Z}, \forall w, v \in \mathcal{W} : |f(w, z) - f(v, z)| \leq \mathrm{L}_0 \|w - v\|$, (2) $f$ is $\mathrm{L}_1$-*smooth* iff there exists $\mathrm{L}_1 \in \mathbb{R}$ such that $\forall z \in \mathcal{Z}, \forall w, v \in \mathcal{W} : \|\nabla f(w, z) - \nabla f(v, z)\| \leq \mathrm{L}_1 \|w - v\|$, (3) $f$ has a $\mathrm{L}_2$-*Lipschitz Hessian* iff there exists $\mathrm{L}_2 \in \mathbb{R}$ such that $\forall z \in \mathcal{Z}, \forall w, v \in \mathcal{W} : \|\nabla^2 f(w, z) - \nabla^2 f(v, z)\| \leq \mathrm{L}_2 \|w - v\|$, (4) $f$ is $\mu$-*strongly convex* iff for all $w, v \in \mathcal{W}$ and $z \in \mathcal{Z}$ we have $f(v, z) \geq f(w, z) + \langle \nabla f(w, z), v - w \rangle + \frac{\mu}{2} \|v - w\|^2$.

### 3.1 Zero-Concentrated DP

For our privacy analysis, we use concentrated differential privacy [DR16; BS16], as it provides a simpler composition theorem – the privacy parameter $\rho$ adds up when we compose.

**Definition 3.1** ([BS16, Def. 1.1])**.** A randomized mechanism $\mathcal{A} : \mathcal{Z}^n \to \mathcal{M}_1(\mathcal{R})$ is $\rho$-zCDP, iff, for every neighbouring dataset (i.e., addition or removal) $S_n \in \mathcal{Z}^n$ and $S_n' \in \mathcal{Z}^n$, and for every $\alpha \in (1, \infty)$, it holds $\mathrm{D}_\alpha(\mathcal{A}(S_n) \| \mathcal{A}(S_n')) \le \rho\alpha$, where $\mathrm{D}_\alpha(\mathcal{A}_n(S_n) \| \mathcal{A}_n(S_n'))$ is the $\alpha$-Renyi divergence between $\mathcal{A}_n(S_n)$ and $\mathcal{A}_n(S_n')$.

We should think of $\rho \approx \varepsilon^2$: to attain $(\varepsilon, \delta)$-DP, it suffices to set $\rho = \frac{\varepsilon^2}{4\log(1/\delta)+4\varepsilon}$ [BS16, Lem. 3.5].

**Lemma 3.2** ([BS16, Prop. 1.3])**.** *Assume we have a randomized mechanism $\mathcal{A} : \mathcal{Z} \to \mathcal{M}_1(\mathcal{R})$ that satisfies $\rho$-zCDP, then for every $\delta > 0$, $\mathcal{A}$ is $(\rho + 2\sqrt{\rho\log(1/\delta)}, \delta)$-DP.*

# 4    Optimal Algorithm for the Class of Strongly Convex Functions

In this section, we present a DP variant of the cubic-regularized Newton's method of Nesterov and Polyak [NP06]. To motivate the idea behind our algorithm, we revisit DP gradient descent (DP-GD) for the class of $\mathrm{L}_0$-Lipschitz and $\mathrm{L}_1$-smooth convex loss functions.

Let $\{w_t^{\text{GD}}\}_{t \in [T]}$ be the iterates of DP-GD. The smoothness of $\ell$ lets us construct a global quadratic upper bound on the function [Nes98, Thm. 2.1.5] as follows $\forall w \in \mathcal{W}$ and $S_n \in \mathcal{Z}^n$ :

$$\ell(w, S_n) \le q_t(w) \triangleq \ell(w_t^{\text{GD}}, S_n) + \left\langle \nabla\ell(w_t^{\text{GD}}, S_n), w - w_t^{\text{GD}} \right\rangle + \frac{\mathrm{L}_1}{2} \left\| w - w_t^{\text{GD}} \right\|^2. \quad (2)$$

Then, DP-GD can be seen as a two-step process:

$$\text{(Step I)} \quad v_{t+1} = \arg\min_v q_t(v) = w_t^{\text{GD}} - \mathrm{L}_1^{-1}\nabla\ell(w_t^{\text{GD}}, S_n), \quad \text{(Step II)} \quad w_{t+1}^{\text{GD}} = \Pi_{\mathcal{W}}(v_{t+1} + \mathrm{L}_1^{-1}\xi_t),$$

where $\xi_t = \mathcal{N}(0, \sigma^2 I_d)$ with $\sigma^2 = \frac{\mathrm{L}_0^2}{2\rho n^2}$ so that $w_{t+1}^{\text{GD}}$ satisfies $\rho$-zCDP [BS16, Lem. 2.5]. That is, in each iteration of DP-GD, *we find a minimum of the quadratic upper bound $q_t(w)$ and then project back to $\mathcal{W}$*. (In the unconstrained setting where $\mathcal{W} = \mathbb{R}^d$ we do not need the second projection step.)

Consider the class of $\mathrm{L}_2$-Lipschitz Hessian convex loss functions. Nesterov and Polyak [NP06, Lem. 1] show that we can construct a *global cubic upper bound* exploiting the second-order information (i.e., Hessian) as follows: for all $w$ and $w_t$, $\ell(w, S_n) \le \phi_t(w)$ where

$$\phi_t(w) \triangleq \ell(w_t, S_n) + \langle \nabla\ell(w_t, S_n), w - w_t \rangle + \frac{1}{2}\langle \nabla^2\ell(w_t, S_n)(w - w_t), w - w_t \rangle + \frac{\mathrm{L}_2}{6}\|w - w_t\|^3. \quad (3)$$

Their non-private algorithm is based on the *exact* minimization of $\phi_t(w)$, i.e., the next iterate is $w_{t+1} = \arg\min \phi_t(w)$. Note that $\arg\min \phi_t(w)$ does not admit a closed form solution, as opposed to the quadratic upper bound (2). Similar to the intuition for DP-GD on smooth loss functions (2), our algorithms in this section are based on *privately* minimizing $\phi_t(w)$ at each iteration. Our algorithm is shown in Algorithm 1. In each iteration the algorithm makes an oracle call to obtain $(\ell(w_t, S_n), \nabla\ell(w_t, S_n), \nabla^2\ell(w_t, S_n))$. Then, the algorithm calls an efficient DPSolver for privately optimizing the cubic upper bound (3). The privacy analysis of Algorithm 1 is a direct application of the composition property of zCDP [BS16, Lemma 2.3]; the output of DPSolver at each iteration satisfies $\rho/T$-zCDP where $\rho$ is the total privacy budget and $T$ is the total number of iterations.

*Remark* 4.1. DPSolver in Algorithm 1 does not affect the *oracle complexity* of Algorithm 1, as it is applied to the proxy loss $\phi_t(w)$, rather than the underlying loss $\ell(w, S_n)$. ◁

---

**Algorithm 1** Meta Algorithm

1: Input: training set $S_n \in \mathcal{Z}^n$, privacy budget $\rho$-zCDP, initialization $w_0 \in \mathcal{W}$, number of iterations $T$.
2: **for** $t = 0, \ldots, T-1$ **do**
3:     Query $\ell(w_t, S_n), \nabla\ell(w_t, S_n), \nabla^2\ell(w_t, S_n)$
4:     Construct $\phi_t(w)$ from Equation (3)
5:     $w_{t+1} = \text{DPSolver}(\phi_t(w), \rho/T, w_t)$
6: Output $w_T$.

---

**Algorithm 2** DPSolver

1: Input: function $\phi : \mathcal{W} \to \mathbb{R} : \phi(\theta) = \ell + \langle g, \theta - \theta_0 \rangle + \frac{1}{2}\langle H(\theta - \theta_0), (\theta - \theta_0) \rangle + \frac{\mathrm{L}_2}{6}\|\theta - \theta_0\|^3$, privacy budget $\tilde{\rho}$-zCDP, initialization $\theta_0$.
2: $N = \frac{2\tilde{\rho}(\mathrm{L}_0 + \mathrm{L}_1 D + \mathrm{L}_2 D^2)^2 n^2}{(\mathrm{L}_0 + \mathrm{L}_1 D)^2 d}, \sigma^2 = \frac{N(\mathrm{L}_0 + \mathrm{L}_1 D)^2}{2\tilde{\rho}}$
3: **for** $i = 0, \ldots, N-1$ **do**
4:     $\eta_i = \frac{2}{\mu(i+2)}$
5:     $\text{grad}_i = g + H(\theta_i - \theta_0) + \frac{\mathrm{L}_2}{2}\|\theta_i - \theta_0\|(\theta_i - \theta_0)$.
6:     $\theta_{i+1} = \Pi_{\mathcal{W}}(\theta_i - \eta_i(\text{grad}_i + \mathcal{N}(0, \sigma^2 I_d)))$
7: Return $\sum_{i=0}^{N-1} \frac{2i}{N(N+1)}\theta_i$

**Theorem 4.2.** *Let $f$ be a $L_0$-Lipschitz, $L_1$-smooth, $L_2$-Lipschitz Hessian, and $\mu$-strongly convex function. Also, assume that $\mathcal{W} \subseteq \mathbb{R}^d$ has finite diameter $D$. Let $w^\star = \arg\min_{w \in \mathcal{W}} \ell(w, S_n)$. Then, for every $\rho > 0$, $\beta \in (0, 1)$, and $S_n \in \mathcal{Z}^n$ for sufficiently large $n$, by setting the number of iterations in Algorithm 1 to*

$$T = \Theta\Big(\frac{\sqrt{L_2}}{\mu^{3/4}}(\ell(w_0, S_n) - \ell(w^\star, S_n))^{\frac{1}{4}} + \log\log\big(\frac{n\sqrt{\rho}}{\sqrt{\log(1/\beta)d}}\big)\Big),$$

*and using Algorithm 2 as* DPSolver*, we have the following: The output of Algorithm 1, i.e., $w_T$, satisfies $\rho$-zCDP and with probability at least $1 - \beta$*

$$\ell(w_T, S_n) - \ell(w^\star, S_n) \leq \tilde{O}\Big(\frac{d(L_0 + L_1 D)^2 \log(1/\beta)}{\mu\rho n^2} \cdot \big(\frac{L_2^2 L_0 D}{\mu^3}\big)^{\frac{1}{4}}\Big)$$

*Remark* 4.3. The lower bound on the excess error of any DP algorithm for the class of strongly convex functions [BST14, Thm. 5.5] implies that the achievable excess error in Theorem 4.2 is *optimal* in terms of the dependence on $d$, $\rho$, and $n$. Also, the oracle complexity of our algorithm is an exponential improvement over the oracle complexity of first-order methods [STU17].   ◁

*Remark* 4.4. The proof of Theorem 4.2 suggests that Algorithm 1 has two phases. First, while $w_t$ is far from $w^\star$, the convergence rate is $1/T^4$. Second, when $w_t$ is close to $w^\star$, the algorithm exhibits the convergence rate of $\exp(\exp(-T))$. Notice that Algorithm 1 is agnostic to this transition in the sense that we do not have an explicit switching step in Algorithm 1 and Algorithm 2. It is also interesting to note that the transition happens when $\|w_t - w^\star\| \leq 3\mu/4L_2$.   ◁

*Remark* 4.5 (Comparison with [ABL21].). In [ABL21, §4], the authors propose a DP variant of Newton's method. Their main idea is to add independent noise *directly* to the Hessian matrix and the gradient vector using the Gaussian mechanism. They also require that *the Hessian be a rank-1 matrix*. The issue with adding noise directly to a full-rank Hessian matrix is that the noise scales with the dimension $d$, which can lead to a suboptimal excess loss. In contrast, our algorithm has a global convergence without placing restrictions on the rank of the Hessian matrix or the initialization.   ◁

*Remark* 4.6. We showed in Theorem 4.2 that our algorithm has an exponentially smaller *oracle complexity* than the first-order methods in terms of the dependence to $n$. For the class of convex, smooth, Lipschitz, and strongly convex, [ZZMW17] proposes a first-order algorithm with an oracle complexity of $T_1 = \Theta\left(\sqrt{L_1}/\sqrt{\mu} + \log(n)\right)$. It is important to note that the *constant* term in $T_1$ differs from our result, making a direct comparison challenging. It is an interesting question to develop a second-order DP algorithm with a smaller oracle complexity than both the algorithms proposed in [ZZMW17] and ours in Algorithm 1.   ◁

*Remark* 4.7. The cubic Newton method has a non-private convergence rate of $T^{-2}$ for the class of convex (but not strongly convex) functions [NP06, Thm. 4]. We leave it as an open question whether there exists a DPSolver such that Algorithm 1 achieves an optimal excess error and oracle complexity for convex functions. However, this can be achieved by a DP variant of the first-order accelerated Nesterov's method [Nes98; NJLS09; GL12]; see Appendix A.2.   ◁

## 5   DP Logistic Regression using Second-Order Information

The main limitation of our cubic Newton's method (Algorithm 1) is that each iteration requires solving a nontrivial subproblem. So, despite low oracle complexity, it is computationally expensive. Moreover, many loss functions, such as logistic loss, are not strongly convex in the unconstrained setting. In this section, we aim to develop a fast second-order algorithm for unconstrained logistic regression avoiding this issue. In many real-world classification tasks, the logistic loss is the loss of choice. The logistic loss is a convex surrogate of the 0-1 loss, and satisfies many regularity conditions that give rise to various practical optimization algorithms [Bac10; Erd15; KSJ18]. Also, note that our results in this section can readily be extended to the class of smooth and convex GLMs.

First, we recall the logistic loss function. Let $d \in \mathbb{N}$ and $\mathcal{Z} = \mathcal{B}^d(1) \times \{-1, 1\}$ be the dimension and data space, where $\mathcal{B}^d(1) = \{x \in \mathbb{R}^d : \|x\| \leq 1\}$ is the unit ball in $\mathbb{R}^d$. Let $f_{\text{LL}} : \mathbb{R}^d \times \mathcal{Z} \to \mathbb{R}$ denote the logistic loss function defined as

$$f_{\text{LL}}(w, (x, y)) = \log(1 + \exp(-y \cdot \langle w, x \rangle)). \tag{4}$$

The gradient and Hessian of $f_{\text{LL}}$ are given by

$$\nabla_w f_{\text{LL}}(w, (x, y)) = \frac{-xy}{1 + \exp(y\langle w, x \rangle)}, \quad \nabla_w^2 f_{\text{LL}}(w, (x, y)) = \frac{xx^\top}{(\exp(-\frac{\langle w, x \rangle}{2}) + \exp(\frac{\langle w, x \rangle}{2}))^2}. \tag{5}$$

Newton's method [BV04, §9.5] is based on successively minimizing a *local* second-order Taylor approximation on the function. Newton's method does not guarantee a global convergence [JT16]; the reason is that the second-order Taylor approximation of the logistic loss can greatly underestimate the function. Next we show that it is possible to obtain a quadratic *global upper bound* on the logistic loss function. We will use this to develop an algorithm that converges globally.

**Lemma 5.1.** *For every $v \in \mathbb{R}^d$, $x \in \mathbb{R}^d$, $w \in \mathbb{R}^d$, and $y \in \{-1, +1\}$, we have*

$$f_{\mathrm{LL}}(w, (x, y)) \leq f_{\mathrm{LL}}(v, (x, y)) + \langle \nabla f_{\mathrm{LL}}(v, (x, y)), w - v \rangle + \frac{1}{2} \langle H_{qu}(v, (x, y))(w - v), w - v \rangle,$$

*where $H_{qu}(v, (x, y)) \triangleq \dfrac{\tanh(\langle x, v \rangle / 2)}{2 \langle x, v \rangle} xx^\top \in \mathbb{R}^{d \times d}$.*

*Remark* 5.2. Since $f_{\mathrm{LL}}$ is $\frac{1}{4}$-smooth, we can construct a simpler global quadratic upper-bound as follows [Nes98, Thm. 2.1.5]: $f_{\mathrm{LL}}(w, (x, y)) \leq f_{\mathrm{LL}}(v, (x, y)) + \langle \nabla f_{\mathrm{LL}}(v, (x, y)), w - v \rangle + \frac{1}{8} \|w - v\|^2$. Lemma 5.1 is tighter than this, since $H_{\mathrm{qu}}(v, (x, y)) \preccurlyeq \frac{1}{4} I_d$; see Appendix B.2. ◁

*Remark* 5.3. The second-order Taylor approximation and our upper bound in Lemma 5.1 both provide a quadratic approximation of the logistic loss. In the remainder of the paper, we write $H(v, (x, y))$ to refer to both $\nabla^2 f_{\mathrm{LL}}(v, (x, y))$ and $H_{\mathrm{qu}}(v, (x, y))$. We refer to $H(v, (x, y))$ as the second-order information (SOI) and to $H_{\mathrm{qu}}$ as *quadratic upperbound* SOI. Finally, notice both $\nabla^2 f_{\mathrm{LL}}(v, (x, y))$ and $H_{\mathrm{qu}}(v, (x, y))$ are PSD rank-1 matrices, with maximum eigenvalue $\leq \frac{1}{4} \|x\|^2 \leq \frac{1}{4}$. ◁

## 5.1 Algorithm Description

We are given a dataset $S_n = ((x_1, y_1), \ldots, (x_n, y_n)) \in (\mathcal{B}^d(1) \times \{-1, +1\})^n$ and we aim to minimize $\ell_{\mathrm{LL}}(w, S_n) \triangleq \frac{1}{n} \sum_{i \in [n]} f_{\mathrm{LL}}(w, (x_i, y_i))$. Our algorithm iteratively minimizes a quadratic approximation of $\ell_{\mathrm{LL}}(w, S_n)$. Consider

$$q_t(w) \triangleq \ell_{\mathrm{LL}}(w_t, S_n) + \langle \nabla \ell_{\mathrm{LL}}(w_t, S_n), w - w_t \rangle + \frac{1}{2} \langle H(w_t, S_n)(w - w_t), (w - w_t) \rangle, \quad (6)$$

where $H(w_t, S_n) \triangleq \frac{1}{n} \sum_{i \in [n]} H(w_t, (x_i, y_i))$. In the non-private setting the next iterate is set to $w_{t+1} = \arg\min_w q_t(w) = w_t - H(w_t, S_n)^{-1} \nabla \ell_{\mathrm{LL}}(w_t, S_n)$. To develop a private variant of Newton's method, we need to characterize the sensitivity of this update rule. Our key observation is that *the directions corresponding to small eigenvalues of $H(w_t, S_n)$ are more sensitive than the directions corresponding to large eigenvalues*. To overcome this issue, we modify the eigenvalues of $H(w_t, S_n)$ to ensure a minimum eigenvalue $\geq \lambda_0$, where $\lambda_0 > 0$ is a carefully chosen constant. We show how to *adaptively* tune $\lambda_0$ in Section 5.2. This procedure yields the desired stability with respect to neighbouring datasets. Formally, the modification operator is defined as follows:

**Definition 5.4.** Let $A \in \mathbb{R}^{d \times d}$ be a positive semi-definite (PSD) matrix and $\lambda_0 \geq 0$. Define

$$\Psi_{\lambda_0}(A, \mathsf{clip}) = \sum_{i=1}^d \max\{\lambda_0, \lambda_i\} u_i u_i^\top, \quad \Psi_{\lambda_0}(A, \mathsf{add}) = \sum_{i=1}^d (\lambda_i + \lambda_0) u_i u_i^\top = A + \lambda_0 I_d.$$

where $A = \sum_{i=1}^d \lambda_i u_i u_i^\top$ is the eigendecomposition of $A$ – i.e., $0 \leq \lambda_1 \leq \cdots \leq \lambda_d$ are the eigenvalues and $u_1, \ldots, u_d \in \mathbb{R}^d$ are the eigenvectors, which satisfy $\forall i \neq j \; \|u_i\| = 1 \wedge \langle u_i, u_j \rangle = 0$.

Algorithm 3 describes our algorithm. First, we state the privacy guarnatee of Algorithm 3 whose proof can be found in Appendices B.3 and B.4.

**Theorem 5.5.** *Assume in Algorithm 3 we choose* add *for the SOI modification. Then, for every training set $S_n \in (\mathbb{R}^d \times \{-1, +1\})^n$, $w_0 \in \mathcal{W}$, $\lambda_0 > 0$, $T \in \mathbb{N}$, $\rho \in \mathbb{R}_+$, and $\theta \in (0, 1)$, by setting $\sigma_1 = \frac{\sqrt{T}}{n \sqrt{2\rho(1-\theta)}}$ and $\sigma_2 = \frac{\sqrt{T}}{(4n\lambda_0^2 + \lambda_0)\sqrt{2\rho\theta}}$, $w_T$ satisfies $\rho$-zCDP.*

**Theorem 5.6.** *Assume in Algorithm 3, we choose* clip *for the SOI modification. Then, for every training set $S_n \in (\mathbb{R}^d \times \{-1, +1\})^n$, $w_0 \in \mathcal{W}$, $\lambda_0 > 0$, $T \in \mathbb{N}$, $\rho \in \mathbb{R}_+$, and $\theta \in (0, 1)$ such that $n > \frac{1}{4\lambda_0}$, by setting $\sigma_1 = \frac{\sqrt{T}}{n \sqrt{2\rho(1-\theta)}}$ and $\sigma_2 = \frac{\sqrt{T}}{(4n\lambda_0^2 - \lambda_0)\sqrt{2\rho\theta}}$, $w_T$ satisfies $\rho$-zCDP.*

**Algorithm 3** Newton Method with Double noise

1: Inputs: training set $S_n \in \mathcal{Z}^n$, $\lambda_0 > 0$, $\theta \in (0,1)$, privacy budget $\rho$-zCDP, initialization $w_0$, number of iterations $T$, SOI modification $\in \{\mathsf{clip}, \mathsf{add}\}$.
2: Set $\sigma_1 = \frac{\sqrt{T}}{n\sqrt{2\rho(1-\theta)}}$
3: **if** SOI modification = Add **then**
4: $\quad \sigma_2 = \frac{\sqrt{T}}{(4n\lambda_0^2 + \lambda_0)\sqrt{2\rho\theta}}$
5: **else if** SOI modification = Clip **then**
6: $\quad \sigma_2 = \frac{\sqrt{T}}{(4n\lambda_0^2 - \lambda_0)\sqrt{2\rho\theta}}$
7: **for** $t = 0, \dots, T-1$ **do**
8: $\quad$ Query $\nabla f(w_t, S_n)$ and $H(w_t, S_n)$
9: $\quad \tilde{H}_t = \Psi_{\lambda_0}(H(w_t, S_n), \text{SOI modification})$
10: $\quad \tilde{g}_t = \nabla f_{\mathrm{LL}}(w_t, S_n) + \mathcal{N}(0, \sigma_1^2 I_d)$
11: $\quad w_{t+1} = w_t - \tilde{H}_t^{-1}\tilde{g}_t + \mathcal{N}(0, \|\tilde{g}_t\|^2 \sigma_2^2 I_d)$
12: Output $w_T$.

Our DP algorithm differs from the non-private Newton's method in three ways: (1) We first privatize the gradient by adding noise. (2) We modify $H(w_t, S_n)$ to ensure its eigenvalues are not too small. And (3) we add a second noise to the update computed using the noised gradient and modified second-order information (SOI).

Notice that Algorithm 3 has *four variations* based on the SOI and the modification of SOI, namely, Hess-clip, Hess-add, QU-clip, and QU-add which refer to using Hessian and clip, Hessian and add, quadratic upper bound (See Lemma 5.1) and clip, and quadratic upper bound and add, respectively.

*Remark* 5.7 (Generalization of Algorithm 3). In this section our main focus is on DP logistic regression, and the privacy guarantees hold for the logistic loss. Nevertheless, in Appendix B.6, we present a generalization of Algorithm 3 whose privacy guarantee holds for *every* convex, doubly differentiable, Lipschitz, and smooth loss function *without any constraints on the rank of Hessian*. The main technical challenge for sensitivity analysis is proving the approximate Lipschitzness of $\Psi$ in the operator norm (See Lemma B.7). This demonstrates that our algorithm is more general than objective perturbation [CMS11; KST12; INST+19] and the private damped Newton's method [ABL21] which both require a low-rank Hessian. ◁

## 5.2 Private and Adaptive Selection of Minimum Eigenvalue

One of the hyperparameters of Algorithm 3 is the minimum eigenvalue $\lambda_0$. There exists a tradeoff for choosing $\lambda_0$. We ideally want the modification to be as small as possible, so that the SOI is preserved. However, decreasing $\lambda_0$ increases $\sigma_2$ and we add more noise. To deal with this problem, we propose a heuristic rule for an adaptive, private, and time-varying selection of the minimum eigenvalue. We wish to find $\lambda_{0,t}$ that minimizes expected loss at the next iteration, for which we have the quadratic approximation (6). More formally, we compute $\lambda_{0,t}$ as $\arg\min_\lambda \mathbb{E}\left[q_t\left(w_t - \Psi_\lambda(H(w_t, S_n), \text{SOI modification})\tilde{g}_t + \|\tilde{g}_t\|\sigma_2(\lambda) \cdot \xi\right)\right]$ where $q_t$ is given in (6) and $\xi \sim \mathcal{N}(0, I_d)$. We show in Appendix B.5 that an approximate minimizer is $\lambda_{0,t} \propto \left(\frac{\text{trace}(H_t(w_t, S_n))}{n^2 \times \text{privacy budget for the direction}}\right)^{\frac{1}{3}}$. Note that $\lambda_{0,t}$ depends on the data through $\text{trace}(H(w_t, S_n))$, which has sensitivity $1/4n$, so it can be estimated privately. In Appendix B.5, we provide the algorithmic description of a variant of Algorithm 3 with an adaptive and private minimum eigenvalue. In particular, we divide the privacy budget at each iteration into three parts: (1) privatizing the gradient; (2) estimating the trace of SOI; and (3) privatizing the direction. We use this variant for our numerical experiments in Section 6.

## 5.3 Convergence Results for Algorithm 3

In this section, we provide data-dependent convergence guarantees for Algorithm 3. We express these guarantees in terms of the conditional expectation $\mathbb{E}_t[\cdot] = \mathbb{E}\left[\cdot|\{w_i\}_{i\in[t]}\right]$ and they can be easily extended to obtain high probability bounds. Before presenting the results, we introduce a notation. For a dataset $S_n = ((x_1, y_1), \dots, (x_n, y_n)) \in (\mathbb{R}^d \times \{-1, +1\})^n$, let $V \in \mathbb{R}^{d \times d}$ denote the *orthogonal projection matrix* on the linear subspace spanned by $\{x_1, \dots, x_n\}$. For every vector $u \in \mathbb{R}^d$, define $\|u\|_V \triangleq \sqrt{u^\top V u}$. This norm naturally arises since for every $w \in \mathbb{R}^d$ we have $\ell_{\mathrm{LL}}(w, S_n) - \ell_{\mathrm{LL}}(w^\star, S_n) \le \frac{1}{8}\|w - w^\star\|_V^2$ where $w^\star = \arg\min \ell_{\mathrm{LL}}(w, S_n)$ (See Appendix B.7).

### 5.3.1 Local Convergence Guarantee of Hess-clip and Hess-add

**Theorem 5.8.** *Let $S_n$ denote the dataset and* rank *denote the dimension of the linar subspace spanned by $\{x_1, \dots, x_n\}$. Let $\lambda_{min,t}$ be the smallest non-zero eigenvalue of $\nabla^2\ell_{\mathrm{LL}}(w_t, S_n)$ and $\rho$ be*

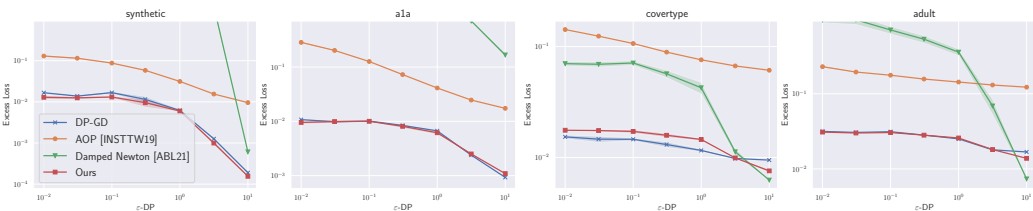

Figure 2: Privacy-Utility tradeoff on different datasets.

*the privacy budget (in zCDP) per iteration. Then,*

$$\mathbb{E}_t \left[ \|w_{t+1} - w^\star\|_V^2 \right] \le \nu_{1,t}^2 \|w_t - w^\star\|_V^2 + 2\nu_{1,t}\nu_{2,t} \|w_t - w^\star\|_V^3 + \nu_{2,t}^2 \|w_t - w^\star\|_V^4 + \Delta,$$

*where the coefficients are given by*

$$\nu_{1,t} = 1 - \frac{\tilde{\lambda}_{min,t}}{\lambda_0} + \frac{\sqrt{\text{rank}}}{(4n\lambda_0^2 - \lambda_0)\sqrt{2\rho\theta}}, \quad \nu_{2,t} = \frac{0.05}{\tilde{\lambda}_{min,t}}, \quad \Delta = O\left(\frac{\text{rank}}{\rho(1-\theta)n^2}\frac{1}{(\tilde{\lambda}_{min,t})^2}\right). \quad (7)$$

*Here,* $\tilde{\lambda}_{min,t} = \begin{cases} \min\{\lambda_{min,t}, \lambda_0\} & \textit{for } \mathsf{Hess\text{-}clip}, \\ \lambda_{min,t} + \lambda_0 & \textit{for } \mathsf{Hess\text{-}add}, \end{cases}$ *depends on the modification procedure.*

This type of convergence is known as *composite convergence*, as it is a combination of linear and quadratic rates, and has been observed in the convergence analysis of several quasi-Newton's methods [EM15; Erd15; RM16; XYRRM16].

*Remark* 5.9. $\lambda_{min,t}$ is the smallest *non-zero* eigenvalue of $\nabla^2 \ell_{\text{LL}}(w_t, S_n)$. Therefore, for sufficiently large $n$ we have $0 < \nu_{1,t} < 1$. It shows Algorithm 3 with Hessian as SOI is, in-expectation, a descent algorithm locally given $\|w_t - w^\star\|$ is sufficiently larger than $\Delta$. Roughly speaking, Theorem 5.8 guarantees a linear convergence to a ball around the optimum whose radius is given by $\Delta$. We also observe the linear rate in Figure 3. Moreover, the error due to the privacy, i.e., $\Delta$ in Equation (7), is proportional to the rank of the feature vectors which is always smaller than $d$. These interesting properties is due to the convergence analysis with respect to $\|\cdot\|_V$. ◁

*Remark* 5.10. The coefficients of the convergence in Equation (7) depend on the iteration step which is an undesirable aspect of the results. In Lemma B.11, we prove that $|\lambda_{min,t} - \lambda_{min}^\star| \le 0.1 \|w_t - w^\star\|_V$ where $\lambda_{min}^\star$ is the smallest non-zero eigenvalue of $\nabla^2 \ell_{\text{LL}}(w^\star, S_n)$. Therefore, the coefficients can be well-approximated by their analogous values evaluated at the optimum. ◁

### 5.3.2 Global Convergence Guarantee of QU-clip and QU-add

We also establish a global convergence guarantee for QU-clip and QU-add. Due to the space the formal statement and proof are deferred to Appendix B.9. Roughly speaking, under the assumption of *local strong convexity at the optimum* [Bac14], QU-clip and QU-add converge globally: this is intuitive since QU-clip and QU-add are based on minimizing a global upper bound on the function.

## 6 Numerical Results

In this section, we evaluate the performance of our algorithm (Algorithm 3 with the adaptive minimum eigenvalue selection from Section 5.2) for the problem of *binary classification* using *logistic regression*. For brevity, many of the details behind our implementation and more experimental results are deferred to Appendix C.

### 6.1 Setup

The setup of the experiments is as follows: **Baseline1- DP-(S)GD**: The update rule is $w_{t+1} = w_t - \eta\nabla\ell(w_t, S_n) + \xi$ where $\xi$ is a Gaussian noise [SCS13; BST14; ACGM+16]. Since the logistic loss is 1-Lipschitz, we do not need gradient clipping. The Lipschitzness parameter controls the variance of the Gaussian random vector. To draw a fair comparison and show the advantage of using second-order information, we chose the stepsize to be equal to the inverse smoothness. **Baseline2-**

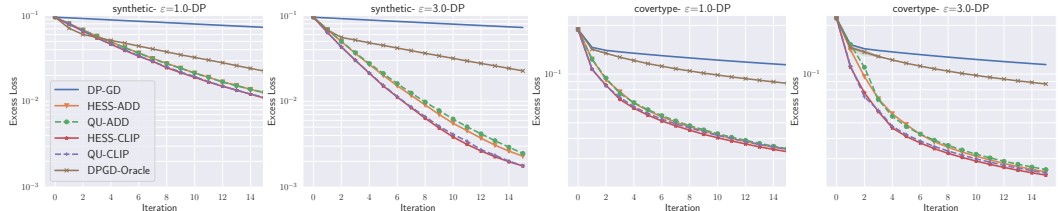

Figure 3: Comparison with DP-GD Oracle where at each iteration the stepsize tuned non-privately.

| | $\dfrac{T^{\star}_{\mathrm{DP\text{-}GD}}}{T^{\star}_{\mathrm{ours}}}$ | | | | $T^{\star}_{\mathrm{ours}}(\mathrm{sec})$ | |
|---|---|---|---|---|---|---|
| | $\varepsilon = 0.01$ | $\varepsilon = 0.1$ | $\varepsilon = 1$ | $\varepsilon = 10$ | $\min(T^{\star}_{\mathrm{ours}})$ (sec.) | $\max(T^{\star}_{\mathrm{ours}})$ (sec.) |
| a1a | $4.87\times$ | $2.95\times$ | $5.09\times$ | $30.59\times$ | 2.45 | 4.2 |
| synthetic | $2.90\times$ | $2.90\times$ | $5.19\times$ | $11.61\times$ | 0.18 | 0.21 |
| adult | $12.08\times$ | $11.84\times$ | $22.17\times$ | $38.16\times$ | 6.81 | 8.07 |
| covertype | $24.19\times$ | $19.85\times$ | $35.70\times$ | $36.20\times$ | 2.93 | 3.58 |

Table 1: Comparison between the run time of our algorithm and DP-GD in terms of the ratio $T^{\star}_{\mathrm{DP\text{-}GD}}/T^{\star}_{\mathrm{our}}$. The last two columns show the minimum and maximum run time of our algorithm.

**Approximate Objective Perturbation (AOP)**: AOP is built on objective perturbation [CMS11; KST12]. Objective perturbation consists of a two-stage process: (1) *perturbing* the objective function by adding a random linear term and (2) outputting the minimum of the perturbed objective. Releasing such a minimum is sufficient for achieving DP guarantees [CMS11; KST12], but only if we can find the exact minimum of the perturbed objective. AOP extends objective perturbation to permit using an *approximate* minimum of the perturbed objective [INST+19; INST+]. Notice AOP is not an iterative optimization algorithm. **Baseline3- Damped Newton Method [ABL21]**: The algorithm in [ABL21] is a variant of damped Newton's method with the assumption that the Hessian of loss function is rank-1, which holds for the logistic loss. Their algorithm is based on adding two i.i.d. noises to the Hessian and the gradient: $w_{t+1} = w_t - \eta_t H_{\mathrm{noisy},t}(w_t, S_n)^{-1}\tilde{g}_t$, where $\eta_t$ is the stepsize, $H_{\mathrm{noisy},t}(w_t, S_n) = \nabla^2 \ell_{\mathrm{LL}}(w_t, S_n) + \Xi_t$ and $\tilde{g}_t = \nabla \ell_{\mathrm{LL}}(w_t, S_n) + \xi_t$. Here $\Xi_t$ and $\xi_t$ are carefully chosen Gaussian noise. With $\eta_t = 1$, our experiments show that their algorithm is not converging. We use the strategy suggested in [ABL21, Page 22] and set $\eta_t = \log(1 + \beta_t)/\beta_t$ where $\beta_t = \left\| \nabla^2 \ell_{\mathrm{LL}}(w_t, S_n)^{-1}\nabla \ell_{\mathrm{LL}}(w_t, S_n) \right\|$. This stepsize selection makes the algorithm *non-private*, however, it serves as a good baseline. **Datasets:** We conducted experiments on six publicly available datasets: a1a, Adult, covertype, synthetic, fashion-MNIST, and protein (Appendix C includes fashion-MNIST and protein results). The synthetic dataset is generated as follows: Fix $d \in \mathbb{N}$ and $w^{\star} \in \mathbb{R}^d$. Then, (1) the feature vectors $\{x_i \in \mathbb{R}^d : i \in [n]\}$ are independent and sampled uniformly at random from the unit sphere in $\mathbb{R}^d$, (2) for the $i$-th datapoint the label is $+1$ with probability $(1 + \exp(-\langle x_i, w^{\star}\rangle))^{-1}$ and $-1$ otherwise. **Privacy Notion:** The privacy notion for our experiments is $(\epsilon, \delta = (\text{num. of samples})^{-2})$-DP. Next, we present the results.

### 6.2 Privacy-Utility-Run Time Tradeoff

We study the tradeoff for our algorithm and compare it with other baselines for a broad range of $\varepsilon \in \{0.01, \ldots, 10\}$. We *non-privately tune* the total number of iterations of the iterative algorithms and report the best achievable excess error in Figure 2. As can be seen our algorithm almost always achieves the best excess loss for a broad range of $\epsilon$. Also, Figure 2 shows that damped private Newton method of [ABL21] achieves a low excess loss only for large $\epsilon$. Figure 2 indicates that DP-GD and our algorithm are the best in terms of excess loss. In Table 1, we compare the run time of DP-GD and our algorithm, i.e., the computational time in seconds for achieving the excess loss in Figure 2. As can be seen, for many challenging datasets, our algorithm is $10$-$40\times$ faster than DP-GD. Our experiments are run on CPU. We also remark that each step of Algorithm 3, i.e., computing gradient and SOI, is heavily parallelizable implying that the run time of Algorithm 3 can be made much smaller by an efficient implementation. Also, the reported numbers in Figure 2 and Table 1 correspond to Hess-clip.

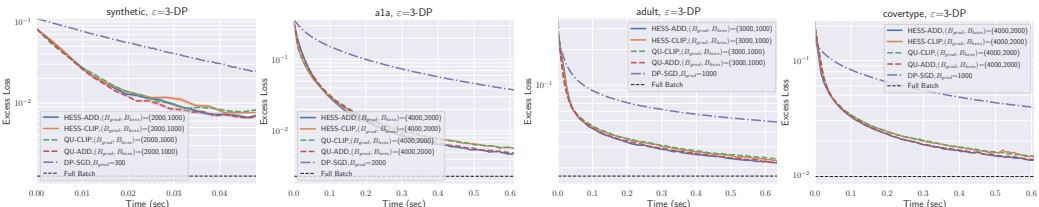

Figure 4: Minibatch Variant of Our Algorithm and Comparison with DP-SGD

### 6.3 Second Order Information vs Optimal Stepsize

In non-private convex optimization, the key to the success of second-order optimization algorithms is that the second-order information acts as a preconditioner, and the same performance *cannot* be attained by optimally tuning the stepsize for GD algorithm. To investigate whether the same holds for our algorithms, we consider the following variant of DP-GD. Let $\tilde{g}_t$ denote the perturbed gradient obtained by adding a Gaussian random vector to $\nabla \ell_{\mathrm{LL}}(w_t, S_n)$. Instead of a constant stepsize, the stepsize at iteration $t$ is chosen based on $\eta_t = \arg\min_{\eta \geq 0} \ell_{\mathrm{LL}}(w_t - \eta \tilde{g}_t)$. Notice this variant is obviously *not DP*. We refer to this variant as *DP-GD-Oracle*. The comparison with DP-GD-Oracle lets us answer the following question: *Could we have just computed a single number, i.e., stepsize, to achieve the same performance as our second-order optimization algorithms which require computing a $d \times d$ matrix?* In Figure 3, we compare the convergence speed of our algorithms with DP-GD-Oracle in low- and high-privacy regimes. Figure 3 shows our algorithms converge faster than DP-GD-Oracle which is not even a DP algorithm. Figure 3 confirms the expectation that as the privacy budget increases the difference between our algorithms and DP-GD-Oracle increases since we can use more curvature information.

### 6.4 Minibatch Variant of Our Algorithm and Comparison with DP-SGD

So far we have considered full-batch algorithms that compute first- and second-order information on the entire dataset. We extend Algorithm 3 to the minibatch setting, where, at each iteration, the gradient and SOI matrix are computed using a subsample of the data points. In Appendix C.1 we provide a formal algorithmic description of the minibatch version of Algorithm 3 along with its privacy proof. Then, we compare the convergence speed and excess loss with DP-SGD.

DP-SGD is faster than DP-GD, but to achieve good privacy and utility, we need large batches [PHKX+23, Fig. 2]. This is in stark contrast with non-private SGD, where larger batch sizes yield diminishing returns [ZLNM+19]. In particular, to achieve the best excess loss we need to select the batch size as large as possible. We select the batch size of DP-SGD so that the achievable excess loss will be close to the full batch versions. Specifically, we select $\frac{\text{batch size DP-SGD}}{\text{number of samples}} \approx 0.02$ and tune the number of iterations of DP-SGD to obtain the best result. Figure 4 shows the progress of different algorithm versus run time. Obviously, for a fixed run time DP-SGD performs more iterations compared to our algorithms. Nevertheless, our algorithms achieve the same excess error as DP-GD with $8$-$10\times$ faster run time over all the datasets while *the batch sizes of our algorithms are larger than that of DP-SGD*. We observe that the variations of our algorithms based on the adding operator performs better in the minibatch setting. This can be attributed to the smaller $\sigma_2$ for the adding operator in Algorithm 3. In summary, the comparison between privacy-utility-wall time tradeoff of the subsampled variant of our algorithm and DP-SGD is similar to their full-batch counterparts.

## 7 Conclusion and Limitations

We showed that second-order methods can be used in the DP setting both for improving worst-case convergence guarantees and designing faster practical algorithms. We believe our results open up many directions: A limitation of our algorithms is that the cost of forming and inverting the Hessian can be prohibitive when $d$ is large. In the non-private setting, a line of research tries to address this limitation by constructing an approximation to SOI such that the update is efficient, yet still provides sufficient SOI [EM15; Erd15; XYRRM16; ABH17]. It would be interesting to investigate how the ideas developed in our paper could be incorporated into these methods.

## Acknowledgments

The authors would like to thank Murat Erdogdu, Jalaj Upadhyay, and Mohammad Yaghini for helpful discussions. Resources used in preparing this research were provided, in part, by the Province of Ontario, the Government of Canada through CIFAR, and companies sponsoring the Vector Institute `www.vectorinstitute.ai/partners`.

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
