# OpenReview forum: "Faster Differentially Private Convex Optimization via Second-Order Methods"
_NeurIPS.cc/2023/Conference — NeurIPS 2023 poster_

### Official Review · Reviewer_Xt6K · 2023-07-03

**Soundness:** 3 good
**Presentation:** 2 fair
**Contribution:** 3 good
**Rating:** 5
**Confidence:** 2

**Summary:**

This paper studies DP convex optimization by using second-order methods. The authors present a private version of the cubic regularized Newton method and prove it faster than the first-order methods under strongly convex case. They also provide efficient second-order method for solving the DP logistic regression problems.

**Strengths:**

This is the very first paper to study the second-order methods for DP convex optimization. The author present algorithms for both general strongly convex functions and logistic regressions which are very novel. The Algorithm 3 for DP logistic regression consider the special structure of the object function which is very interesting.

**Weaknesses:**

The algorithm for solving the general strongly convex functions seems straight forward. It is a combination of cubic-regularized Newton and DPGD. The requirement of solving subproblem by DPGD at each iteration is not satisfying and may cause the total computation cost even worse than the first-order method.  Can the author present the total number of the iterations (including the inner loop) and compare it with the first-order DP method?

Although the author present efficient method for solving the DP logistic regression, Algorithm 3 seems quite different from the meta algorithm for the general strongly convex optimization problems. Can the authors show the connection between the Algorithm 3 and Algorithm1,2?



**Questions:**

See weakness part.

---

> ### Author Rebuttal · Authors · 2023-08-09
>
> We thank the reviewer for their comments on our submission. We respond to the weaknesses raised.
>
> > **W1**:  The algorithm for solving the general strongly convex functions seems straight forward:
>
> We respectfully disagree. The most natural way to privatize the non-private Newton’s method is to add noise directly to the gradient and Hessian, as was proposed by [ABL21]. However, to achieve optimal excess error with this method we need an additional assumption that the Hessian of the loss function is a low rank matrix; our algorithm does not suffer from such a limitation. Please refer to Remark 4.5 for further discussion. On the practical side we tried many noise adding schemes before settling on the double noise method (Algorithm 3) and we found that this significantly outperformed the simpler approach of adding noise directly to the Hessian matrix.
>
> > **W2** Algorithm for strongly convex functions and its total iteration cost:
>
> We discuss this in [our common response](https://openreview.net/forum?id=h2lkx9SQCD&noteId=ZhStwLJeHt). The total iteration cost depends on the subproblem solver. In particular, as discussed in our common response, the total iteration complexity is $\log\log(n) * \log(n)$. Therefore, the iteration cost is competitive with first-order methods. The win is in the oracle complexity.
>
> > **W3**: Connection between Algorithm 1 and Algorithm 3 in the paper:
>
> Logistic loss is *not strongly convex* in the unconstrained setting. Also, the main limitation of the proposed cubic Newton's method is that each iteration requires solving a nontrivial subproblem. These are the main reasons why we develop a new algorithm for private logistic regression with significantly improved *wall clock time* (in seconds) compared to other baselines, as shown in Table 1. Note that our proposed algorithm is not limited to the logistic loss; we provide a generalization of Algorithm 3 in Appendix C.6. Please see Remark 5.5.

---

> > ### Comment · Reviewer_Xt6K · 2023-08-18
> >
> > Thanks for your detailed reply and I would like to keep my score unchanged.

---

> > > ### Author Response · Authors · 2023-08-18
> > >
> > > Thanks for your reply! We will revise our paper and incorporate your constructive comments.

---

### Official Review · Reviewer_Pzhc · 2023-07-05

**Soundness:** 2 fair
**Presentation:** 2 fair
**Contribution:** 3 good
**Rating:** 5
**Confidence:** 4

**Summary:**

This work investigates the use of second-order methods in differentially private convex optimization for machine learning. The authors propose a private variant of the regularized cubic Newton method and demonstrate its quadratic convergence and optimal excess loss for strongly convex loss functions. They also design a practical second-order differentially private algorithm for unconstrained logistic regression, which outperforms other baselines in terms of excess loss and is significantly faster than DP-GD/DP-SGD, achieving a speedup of 10-40 times.

**Strengths:**

This paper is good as it has been a really hard to use second-order information while preserving DP property. The contribution is novel and the proof seems to be correct.

**Weaknesses:**

1. Maybe extend to use laplace noise?
2. The presentation of the paper is a little bit awkward. For example, the formal DP theorems and proof of Alg. 3 only presents in Appendix. It is really hard to find the corresponding reference. This paper might need to be rewritten.
3. Experiments:
    1. Compare the true running time. It takes a lot of time to compute the CLIP and ADD operations as it needs to compute SVD. So instead of seeing how many steps it needs, please present the true computational time to show the true running time improvement compared with DP-SGD and DP-GD.
    2. Compare memory usage. SVD needs tons of memory.
    3. DP-SGD brings randomness and sometimes accelerates the training procedure. Please add a comparison with DP-SGD.
    4. Compare with different learning rates (I know there does not exist a learning rate in Alg. 3 but a learning rate exists in DP-(S)GD). It would be interesting to see the impact of different learning rates (10, 0.1, or 0.01).

**Questions:**

See Weakness.

**Limitations:**

See Weakness.

---

> ### Author Rebuttal · Authors · 2023-08-09
>
> We thank the reviewer for their thorough review. We respond to the weaknesses and questions below.
>
> > **W1**: Laplace noise and pure DP:
>
> We can potentially use laplace noise to design algorithms with stronger pure-DP guarantee. However, it has been observed empirically and theoretically that the excess loss of convex optimization under pure-DP is much higher than the relaxed notions such as zero-concentrated DP (zCDP). For instance, for DP convex optimization, by Theorem 3.2 and Theorem 2.4 in [BST14] we can see that pure-DP algorithms exhibit an excess loss dependence of O(dimension), while zCDP algorithms demonstrate a dependency of sqrt(dimension).
>
> [BST14] Bassily, Raef, Adam Smith, and Abhradeep Thakurta. "Private empirical risk minimization: Efficient algorithms and tight error bounds." 2014 IEEE 55th annual symposium on foundations of computer science. IEEE, 2014.
>
> > **W2**: Presentation of the paper:
>
> Thanks for the suggestion! We will state the privacy proof more formally as a theorem in the main body of the paper. We would greatly appreciate any other suggestions for improving the presentation.
>
> > **W3-Part1**: Comparison of the true runtime:
>
> In Section 6, we have already compared the runtime (in sec) of our algorithm with DP-(S)GD. Please refer to Table 1 and Fig.2 in the main body and Table 4 in the appendix. For instance, for Covertype dataset our algorithm is 30 times faster compared to DPGD for achieving $\varepsilon=1$. This is remarkable since the per-iteration cost of the second-order methods is higher than DP-(S)GD.
>
> > **W3-Part2**: Comparison of memory usage:
>
> It is correct that SVD may need large memory space. The memory usage varies with the chosen SVD implementation; we utilized numpy's linalg.eigh in our experiments. To address this concern, we'll provide memory usage specifics for SVD on each dataset in the revised paper.  For instance, for Adult dataset  with dimension 100, the memory usage is 52 MiB, and for Fashion-MNIST with dimension 784, the memory usage is 67 MiB.
>
> > **W3-Part3**: Comparison with DP-SGD:
>
> In Section 6.1, we have already compared the minibatch variant of our algorithm with DP-SGD and show that it has faster convergence. Fig. 4 summarizes the results. The subsampled variant of our algorithm achieves the same excess error as DP-SGD with $8$-$10 \times $ faster run time over all the datasets while the batch sizes of our algorithms are larger than that of DP-SGD (Please see Section 6.1).
>
> > **W3-Part4**: Different Learning Rate for DPGD:
>
> Thanks for the suggestion. The **attached PDF (Item 1)** in [our common rebuttal](https://openreview.net/forum?id=h2lkx9SQCD&noteId=ZhStwLJeHt) shows the results for different learning rates for Adult dataset. We will include the results for all the datasets in the next revision of the paper.
> The main observation is that a larger learning rate helps DPGD in the initial phase of optimization. However, after getting close to the optimal point, large learning rate leads to a higher excess error.  Also, in the submitted paper Line 331, we have provided numerical results comparing our algorithm with DP-GD where at each iteration, we perform linesearch to select learning rate. We call this variant DP-GD-Oracle and obviously this variant does not satisfy DP. Nevertheless, Figure 3 shows our algorithms converge faster than DP-GD-Oracle which is not even a DP algorithm.

---

### Official Review · Reviewer_FNjC · 2023-07-06

**Soundness:** 3 good
**Presentation:** 3 good
**Contribution:** 3 good
**Rating:** 5
**Confidence:** 3

**Summary:**

The paper introduces the use of second-order methods in differentially private optimization. In particular, two algorithms are proposed. One is based on cubic regularized Newton and works for strongly-convex functions. The other one is designed specifically for the logistic regression problem. Numerical experiments on the logistic regression are provided.

**Strengths:**

The topic of using second-order methods in differentially private (DP) optimization is under-explored. In this sense, the paper can be a good starting point and introduce the idea to the DP community. The paper is well-written and easy to follow, with theoretical results that do not exist before and are supported by numerical experiments.

**Weaknesses:**

1. The theoretical justification of the benefit of using second-order methods is not strong enough. For the considered smooth strongly-convex setting, first-order methods can already achieve the same rate in linear time $O(n)$ [arXiv:1703.09947, arXiv:1802.05251, arXiv:2005.04763, arXiv:2102.05855, arXiv:2206.00363]. As a comparison, since the subproblem of the cubic regularized Newton cannot be efficiently solved due to nonsmoothness, the total gradient complexity of the proposed method is $NT\sim n^2$. The main improvement is mostly on the oracle complexity but with more assumptions on the second-order information. Even for the oracle complexity, the proposed algorithm achieves $\sqrt{L_2}/\mu^{3/4}+\log\log(n)$, which is hard to compare with the one achieved by first-order methods (e.g., $(\sqrt{L_1}/\sqrt{\mu})\log(n)$ for private versions of the Nesterov's accelerated gradient descent [arXiv:2102.04704, arXiv:2206.00363]). I am also wondering if an improved lower-bound is possible with second-order information.

2. The convergence analysis of the logistic regression case seems to be not complete. I don't find (also in the appendix) any specific rate, complexity, and comparison with first-order methods. The proposed method requires the local strong convexity assumption at the optimum. Does it hold for the logistic regression loss?

3. The proposed method requires computing the inverse of the second-order information, which could be hard for large-scale experiments. Also, all the numerical experiments in this paper use small models and datasets, where computing this inverse does not cost too much. Given that DP-GD already completes the task within 1 minute, such a 10-40$\times$ improvement is not considered to be so surprising.

4. What does $T^*$ mean in Table 1, or what is the stopping criterion to define $T^*$? Why do Figure 3 & 4 only show the first 10 steps or 0.6s? I guess the noise added for the proposed method is computed according to the theory. Are there any numerical privacy accounting methods used in the experiments to justify the effectiveness of the added noises? In case the privacy analysis is wrong and less noise is added than what is required, this might not be a fair comparison.

5. Minors: It might be good to also summarize related works for non-DP second-order optimization algorithms. What does privacy budget for direction mean in $\lambda_{0,t}$ in line 249.



**Questions:**

See weakness.

**Limitations:**

See weakness.

---

> ### Author Rebuttal · Authors · 2023-08-10
>
> We thank the reviewer for their thorough review. We respond to the weaknesses raised below.
>
> >**W1**: Justification of using second-order methods and comparison with first order methods
>
> Our goal in this paper is to understand whether second-order methods are compatible with DP. For the theory, we use oracle complexity, which is the standard measure of convergence rate in the non-private literature.
>
> The subproblem we solve is a cubic function, which is indeed smooth *over the constraint set.* This observation lets us use more efficient algorithms as the subproblem solver. Please refer to [our common response](https://openreview.net/forum?id=h2lkx9SQCD&noteId=ZhStwLJeHt) where we discuss it further. In particular, we can show that the number of gradient and Hessian-vector evaluations is $\log\log(n)* n * \log(n)$.
>
> As mentioned, the oracle complexity of first order methods is $\log n$ while ours is $\log\log(n)$. I.e. our is an exponential improvement in oracle complexity. This result tight, including the other terms, as shown by [non-private oracle complexity lower bounds](https://arxiv.org/abs/1705.07260).
>
> > **W2**: local strong convexity at the optimum for logistic loss and its convergence analysis
>
> Unless the data is linearly separable, the logistic loss has a minimum eigenvalue at the optimum which is ** bounded away from zero**. Let $f(x) = \log(1+\exp(-x))$, then we have $f’’(x)\geq 1/4  \exp(-|x|)$. Therefore, by Eq.(5) in the paper, eigenvalues at the optimum are strictly larger than zero for all eigenvectors that are not orthogonal to the subspace spanned by the data. This assumption has been used before as well. See arxiv:1303.6149.
>
> In the submitted paper, we only present the recursion of the error. It can be easily used to obtain the following for our local convergence result: Assume that the initial point is sufficiently close to the optimum. Assume that $\lambda_0 > \lambda^\star_{min}$, then after $T=\tilde{O}\left( \frac{\lambda_0}{\lambda^\star_{min}} \log(n) \right)$ iterations, Algorithm 3 achieves the excess loss of $\tilde{O}\left( \kappa^\star \frac{\text{rank}}{\rho n^2 \lambda^\star_{min}} \right)$ where $\kappa^\star$ is the condition number at the optimum.
> We will include this result in the paper. Note that the dependence on the condition number and the assumption of the local strong convexity at optimum appear in many prior work on second-order optimization (See arxiv:1508.02810,arxiv:1607.00559)
>
> > **W3**: On significance of speed-up of our algorithm and the choice of datasets
>
> For larger datasets (i.e. larger $n$), the gap between our algorithm and DP-(S)GD increases. For instance, for Adult the completion time for DPGD is 5 minutes while for ours is 8 seconds.  We conducted an additional larger scale experiment for the synthetic dataset with a five times more samples. In the attached PDF Item 2 in our common response, we plot the excess loss versus runtime, and as can be seen again there is a significant gap between the completion time of our algorithm and DPGD.
>
> It is a well-known limitation of second-order methods that they require matrix computations, rather than just vector computations like first-order methods, which is a problem when the dimension $d$ is large. There is an extensive line of work on addressing this challenge by working with low-rank approximations to the second-order information. It would be interesting to combine our methods with these approaches to scale to high-dimensional settings. However, this is beyond the scope of our work. Our goal is simply to demonstrate the feasibility of using second-order information to accelerate DP convex optimization.
>
> We use standard datasets for classification for which the linear classifier learned via logistic regression is successful. The dimension of the datasets are in the range of 55 to 784. We would appreciate the reviewer’s suggestions for further datasets to consider.
>
>
> > **W4**: What does T^\star mean? And what is the stopping criterion?
>
> $T^*$ is the runtime of the algorithm in seconds. The star refers to the fact that we perform hyperparameter tuning to optimize the accuracy and report the runtime for this setting of hyperparameters. Note that we perform tuning of $T^*$ for **all** iterative algorithms.
> Note that the number of steps $T$ is specified a priori as a hyperparameter; it is not a dynamic stopping criterion. This is necessary, as we must know $T$ in order to divide the privacy budget between iterations.
>
>
> > **W4**: Why do Figure 3&4 only show the first 10 steps or 0.6s?
>
> In Figure 3, our goal is not to plot the excess error vs runtime. Our goal was to compare the impact optimal step size for DPGD and second-order information on the excess error. The choice of 15 steps is arbitrary.
>
> Figure 4 provides a comparison between the minibatch variations of our algorithm and DP-SGD. The x-axis capped at our algorithms' maximum $T^\star$ value. Notice that for synthetic data  $T^\star$ of our algorithm is around 0.05. Please see Item (3) in the attached pdf in [our common response](https://openreview.net/forum?id=h2lkx9SQCD&noteId=ZhStwLJeHt) for a plot without truncating x-axis.
>
> > **W4**: Privacy accounting for the experiments
>
> We use the same privacy accountant for all studied algorithms. The full-batch variant of our algorithms and DP-GD satisfies zCDP. zCDP provides a simple composition theorem–the privacy parameter adds up when we compose and it is tight. For translation from DP to zCDP, and for the mini batch variants, we use Opacus package.
>
>
> > **Minor Comments**:
>
> Thanks! We will include a comprehensive literature review of the non-private second-order methods.
>
> For the adaptive scheme we divide the privacy budget into three parts. We refer to the privacy budget for estimation of $\Phi(H)^{-1} (\tilde{g})$ as the privacy budget for the direction.

---

> > ### Comment · Reviewer_FNjC · 2023-08-14
> >
> > Thanks for your reply. I still have some concerns that the authors do not clearly address.
> >
> > > For the oracle complexity, the proposed algorithm achieves $(\sqrt{L_2}/\mu^{3/4})(\ell_0 - \ell^*)^{1/4} + \log\log(n)$, which is hard to compare with the $\sqrt{L_1/\mu}\log(n)$ complexity achieved by first-order methods.
> >
> > 1. I agree that the proposed second-order method achieves exponential speed-up in terms of dependence on $n$. However, the constant $L_2$ and $\ell_0-\ell^*$ can be pretty large in practice. Also, it is hard to compare because of different assumptions and constants.
> >
> > > I am also wondering if an improved lower-bound is possible with second-order information.
> >
> > 2. Could the authors also comment on the lower-bound?
> >
> > > Reply to W1: The subproblem we solve is a cubic function, which is indeed smooth over the constraint set.
> >
> > 3. Then the smoothness parameter depends on the diameter, which could be very large and will enter the number of gradient and Hessian-vector evaluations. In comparison, first-order methods do not necessarily have a dependence on the diameter in their gradient complexity (considering output perturbation for smooth strongly-convex functions).
> >
> > 4. My last concern is regarding the extension to other settings. Since the understanding of the convergence guarantees on second-order methods is restricted to smooth convex functions even for non-DP optimization, the extension to more practical nonconvex problems could be hard. It only looks promising to first run a first-order method to a small neighborhood of local minimum that potentially satisfies the smoothness and strong-convexity assumptions, and then use a second-order method. However, the existence of DP noise might prevent outputs of first-order methods to be in a small neighborhood of local minimum, unless other structural assumptions are made.
> >
> > Except for these, my other concerns are successfully addressed by the author's rebuttal.

---

> > > ### Author Response · Authors · 2023-08-14
> > > **Reply to Reviewer FNjC**
> > >
> > > Thanks again for your detailed comments. We respond to the remaining points.
> > >
> > > 1. Indeed, it is hard to compare Theorem 4.2 directly with first-order methods. As mentioned in Remark 4.4, the analysis suggests a phase transition. When we are far from the optimum, second order information does not help much, but once we are close we achieve an exponential speed up. It is possible that in practice the first phase is more important, but in general the optimization theory literature focuses on the asymptotic convergence rate which is determined by the latter phase. We have tried to address questions about practical performance with our experimental results.
> > >
> > > 2. We haven't thought about lower bounds for second order methods with DP. There are lower bounds for second-order methods without DP and there are information theoretic lower bounds for DP. Other than taking the max of these lower bounds, we don't know what to do here.
> > >
> > > 3. Note that assuming the loss function is Lipschitz and strongly convex implies a bound on the diameter. So a diameter bound is often implicit in the analysis of first order methods even if not explicitly stated.
> > >
> > > 4. Applying second order methods to practical non-convex optimization is a challenge even without privacy, although there is [work on this](https://arxiv.org/abs/2002.09018).
> > >
> > > The main message of our work is that second order methods can work with DP to accelerate optimization. This goes against the commonly held belief that second order methods are too brittle to work with noise. Of course, second order methods have other limitations unrelated to privacy and we do not escape these.
> > >
> > > We hope that the reviewer is convinced of our main message and is willing to support our submission.

---

> > > > ### Comment · Reviewer_FNjC · 2023-08-15
> > > >
> > > > Thanks for the reponse. For the answer 3, the diameter dependence of first-order methods is only logarithmic, i.e., $(L_1/\mu)\log D$ where $L_1$ is the smoothness parameter and $\mu$ is the strong-convexity parameter. As an example, you can find it in Theorem 2 of [arXiv:1703.09947]. At the same time, since the smoothness parameter of the cubic-regularized function depends on the diameter, the diameter dependence of second-order methods can be polynomial. The authors' reply does not clearly address this point.

---

> > > > > ### Author Response · Authors · 2023-08-15
> > > > > **Reply to Reviewer FNjC Regarding Diameter**
> > > > >
> > > > > Note that we can run a first order-method for a few iterations to get close to the optimum and then switch to our second-order method. This would get the best of both worlds. We did not include this, since it would complicate the algorithm without offering any real insight. However, we will add a discussion of this in the next revision.
> > > > >
> > > > > **More detailed explanation of the performance of first-order methods:** It is not as simple as saying that first-order methods have a logarithmic dependence on the diameter, because this is only true under strong assumptions that imply that the diameter cannot be large.
> > > > >
> > > > > Specifically, [Zhang et al.](https://arxiv.org/abs/1703.09947)'s Theorem 2 assumes the objective function is $\mu$-strongly convex and $L$-Lipschitz. This implies an upper bound on the diameter $D \le 2L/\mu$. The oracle complexity of their result (part 2) is $$T = \Theta\left(\frac{\mu^2+\beta^2}{\mu\beta} \log\left(\frac{\mu^2n^2\varepsilon^2D^2}{L^2 d \log(1/\delta)}\right)\right).$$
> > > > > If $D = \Theta(L/\mu)$ (i.e. the diameter is as large as possible), then this cancels to $T = \Theta\left(\frac{\mu^2+\beta^2}{\mu\beta} \log\left(\frac{n^2\varepsilon^2}{ \log(1/\delta)}\right)\right).$
> > > > > To shave the log factor from $T$, we need $D = O \left( \frac{L \sqrt{d} \log(1/\delta)}{\mu n \varepsilon} \right)$.
> > > > > That is to say, their dependence on the diameter is only interesting when the diameter is very small.
> > > > >
> > > > > But small diameter is also the general regime where our Algorithm performs well. Thus, at least in this case, the dependence on the diameter does not appear to be a significant advantage of first-order methods. That said, the results are not directly comparable, so there will be parameter regimes where the result of Zhang et al. beats ours; in that case, we can always combine the algorithms. We can use a first-order method initially and then, once we are close enough, we can switch to our second-order method.
> > > > >
> > > > > We hope that this addresses the reviewer's remaining concerns.

---

> > > > > > ### Comment · Reviewer_FNjC · 2023-08-15
> > > > > >
> > > > > > Thanks for your reply. I agree this could be subtle. I am willing to support the submission, as long as the authors mention these limitations in their work. I have raised my score. Thanks again for the discussion.

---

> > > > > > > ### Author Response · Authors · 2023-08-18
> > > > > > >
> > > > > > > Thanks for your kind reply! We will revise our paper and incorporate your constructive reviews.

---

### Official Review · Reviewer_QyKz · 2023-07-23

**Soundness:** 3 good
**Presentation:** 3 good
**Contribution:** 3 good
**Rating:** 7
**Confidence:** 3

**Summary:**

This paper considers the problem of differentially-private convex optimization and discusses how second-order information can be utilized to accelerate the optimization process while also achieving the optimal excess error, with DP involved. The main contributions of this paper are the two proposed algorithms, and the corresponding analyses; 1) A second-order DP algorithm that is based on the cubic-regularized Newton's method - for a specific class of convex functions, 2) A second-order DP algorithm for the logistic regression problem.

**Strengths:**

The paper considers an important problem which studies how the performance of a differentially-private convex optimization process can be enhanced with the use of second-order information. The paper is well organized and rigorous proofs have been provided for the claims made. The first algorithm presented in this paper (DP variant of the cubic-regularized Newton's Method) achieves the optimal excess loss and quadratic convergence. The proposed second-order DP algorithm for logistic regression achieves equal or better excess losses and lower computational times compared to the existing DP algorithms, based on the experimental results.

**Weaknesses:**

The authors could clearly state their contributions and remark on the significance of the contributions, as in some cases (for example algorithms 1 and 2), it seems like the authors have simply incorporated DP into existing non-private results. The main paper lacks justification on the privacy guarantees. The authors could comment on how the stated DP guarantees are achieved by the selected noise parameters.

**Questions:**

1) For a given set of inputs in algorithm 3, how can one determine whether the SOI modification is "add" or "clip"? The reader could benefit from a clear description of this part (lines 3-6) in algorithm 3.
2) Do the authors assume any characteristics (i.e., any specific distribution or any statistical characteristics) on the dataset $S_n$ in either of the algorithms?

**Limitations:**

The authors have clearly stated the limited types of functions which the proposed algorithms can be applied on, and have also commented on possible extensions to other classes of functions.

---

> ### Author Rebuttal · Authors · 2023-08-09
>
> We thank the reviewer for their thorough review. We respond to the weaknesses and questions below.
>
> > **W1**: The authors could clearly state their contributions and remark on the significance:
>
> The existing DP optimization literature is almost exclusively restricted to first-order (and zero-order) methods. Our goal in this paper is to answer the following question: “Can second-order information yield faster convergence in the differentially private setting?” Combining second-order methods with differential privacy requires a great deal of care, as they are fairly sensitive to noise.  We believe that our results provide a convincing affirmative answer to this question both in terms of worst-case convergence guarantee and practical algorithm design.
>
> > **W2**: it seems like the authors have simply incorporated DP into existing non-private results:
>
> We respectfully disagree. The most natural way to privatize the non-private Newton’s method is to add noise *directly* to the gradient and Hessian, as was proposed by [ABL21]. However, to achieve optimal excess error with this method we need an additional assumption that the Hessian of the loss function is a low rank matrix; our algorithm does not suffer from such a limitation. Please refer to Remark 4.5 for further discussion.
> On the practical side we tried many noise adding schemes before settling on the double noise method (Algorithm 3) and we found that this significantly outperformed the simpler approach of adding noise directly to the Hessian matrix.
>
> > **W3**: Proof of Privacy:
>
> The detailed privacy guarantee of Algorithm 3 is stated and proved in Appendix C.3 and C.4. Based on your comment, we will move the formal statement of privacy guarantee to the main body.
>
> > **Q1**: On “add” or “clip” for second order modification:
>
> We apologize if the presentation of Algorithm 3 caused confusion. The type of second-order information modification is an *hyperparameter* of our proposed algorithm. Also, notice that the modification based on “add” and “clip” have a different privacy proof as can be seen by the scale of noise. Empirically, we observe that for the full-batch setting “clip” is better than “add” and for minibatch version “add” performs better. Based on your suggestion, we have included a more precise discussion of it in the revised version.
>
> > **Q2**: Dataset assumptions.
>
> No. We assume **worst-case** dataset for both our privacy proof and for our convergence guarantees. It is an interesting direction to analyze the generalization performance of our algorithms with distributional assumptions. We will add your suggestion to the list of future work.

---

> > ### Comment · Reviewer_QyKz · 2023-08-12
> >
> > Thank you for the detailed explanations.
> >
> > Regarding W2: I understand that the double noise method introduced in Algorithm 3 which privatizes the gradient and the direction (first and second order information), is different from the typical DP-Newton's method that the authors have described. However, it is not very clear how Algorithm 2 stands out from the typical case. While remark 4.5 explains the fact that Algorithm 2 does not impose any restrictions on the Hessian matrix unlike in the typical case, it would be more clear to the reader if the authors can explain the reason/intuition behind this statement.

---

> > > ### Author Response · Authors · 2023-08-14
> > > **Reply to Reviewer QyKz regarding the intuition behind Algorithm 1**
> > >
> > > We thank the reviewer for their reply.
> > >
> > > The main idea of our result for strongly convex function revolves around using the cubic upperbound and showing that obtaining an **approximate** solution of the cubic subproblems suffices to obtain an optimal excess loss. Notice that the accuracy of the cubic subproblem solver is influenced by the sensitivity of the Hessian, which characterizes the noise scale. In Algorithm 2, we only need the gradient of the cubic function (See Line 5 in Alg.2.). The gradient of the cubic function depends on the Hessian through $H(\theta_i - \theta_0)$, and the scale of the noise depends on the $L_2$ norm of $\|\|H(\theta_i - \theta_0)\|\|\leq M * D$, where $M$ is the smoothness parameter and $D$ is the diameter of the space.
> > >
> > > This approach is different from the straightforward approach where two independent noises are added directly to Hessian and the gradient. The noise for privatizing the Hessian matrix scales with its Frobenius norm. Therefore, unless the rank is bounded, the scale of the noise for Hessian scales with dimension which results in a suboptimal utility bound (See [ABL21, Appendix E].).
> > >
> > > In summary, as opposed to the straightforward approach where gradient and Hessian are privatized independently, we only add noise once since the performance of our algorithm depends on the accuracy of the cubic subproblem solver.

---

> > > > ### Comment · Reviewer_QyKz · 2023-08-16
> > > >
> > > > Thanks for the response. All my comments are addressed at this point.

---

> > > > > ### Author Response · Authors · 2023-08-18
> > > > >
> > > > > Thank you very much for your reply. We will revise our paper according to the constructive comments in the reviews.

---

### Official Review · Reviewer_rkzh · 2023-07-23

**Soundness:** 3 good
**Presentation:** 3 good
**Contribution:** 3 good
**Rating:** 6
**Confidence:** 4

**Summary:**

The paper introduces an algorithm that utilizes second-order information to enhance the speed of private convex optimization, concurrently ensuring optimization excess error is minimal.

The outcomes of the experiments indicate that employing second-order information can expedite Differential Privacy (DP) optimization. In addition, it achieves an excess loss that either equals or surpasses that of first-order methods like DP Gradient Descent (DP-GD).

**Strengths:**

The strength of this paper are

1) The application of second-order methods to convex optimization is challenging research area. While some progress has been made, it's still uncertain whether second-order methods can be as practical as first-order methods. This paper unveils a novel second-order approach for Differential Privacy (DP) optimization, which demonstrates optimal efficiency for strongly convex functions.

2) The authors offer an analysis of both the local convergence assurance associated with Hess-clip and Hess-add, as well as the global convergence guarantee for QU-clip and QU-add.

3) The numerical findings indicate that the proposed method outperforms DP Gradient Descent (DP-GD) substantially, demonstrating a speed that is 10 to 40 times quicker for the datasets tested.

**Weaknesses:**

The weakness the paper include
1) The potential impact of this work remains uncertain to me. While it presents strong results in the field of Differential Privacy (DP) private machine learning, it raises the question: can these results or concepts be applied more broadly?

2) The mini-batch version offers interesting numerical results, however, the loss it produces is notably higher compared to the full-batch version.



**Questions:**

On page 1, "One of the major drawbacks of DP-(S)GD is slow convergence. We argue that the main reason for this is the difficulty of choosing the hyperparameters (/eta; T)." Please explain why it is that. How about gradients?

For minibatch version, how to make sure the second-order information is still relevant/meaningful from one batch to another batch?


**Limitations:**

Yes, I think so.

---

> ### Author Rebuttal · Authors · 2023-08-09
>
> We thank the reviewer for their thorough review. We respond to the weaknesses and questions below.
>
> > **W1** potential impact
>
> DP optimization is an important area with numerous practical applications. Our results are a substantial deviation from the existing DP optimization literature, which is almost exclusively restricted to first-order (and zero-order) methods. We believe our work can open up many directions for future investigation within the area of DP optimization. Beyond DP, we speculate that our results may be helpful for designing second-order convex optimization algorithms under **data corruptions**, i.e., robust second-order optimization. Recently, it has been a flurry of interest in understanding the connection between robust algorithms and private algorithms. (see [Asi+23].)
>
> [Asi+23] Asi, Hilal, Jonathan Ullman, and Lydia Zakynthinou. ["From robustness to privacy and back."](https://arxiv.org/abs/2302.01855) (2023).
>
> > **W2** weaker minibatch results:
>
> This is an important observation and one that has been observed repeatedly in the differential privacy literature. In general, it has been observed that for DP optimization the best results are attained by larger  batch sizes. E.g., see Figure 1 of [P+23] which shows that for training a neural network with DP, the required batch size needs to be large in order to reduce the amount of noise. It is an interesting phenomena that we have also observed in our experiments. It is an open question to show that such a limitation is inherent.
>
> [P+23] Ponomareva, Natalia, et al. ["How to dp-fy ml: A practical guide to machine learning with differential privacy."](https://arxiv.org/abs/2303.00654) Journal of Artificial Intelligence Research 77 (2023).
>
> > **Q1a**: Why is it difficult to set hyperparameters for DP-(S)GD?
>
> Please see [our common rebuttal](https://openreview.net/forum?id=h2lkx9SQCD&noteId=ZhStwLJeHt) elaborating on this point. We will try to clarify this important point in the revision.
>
> > **Q1b**: How to make sure second-order information is still relevant/meaningful from one batch to another batch?
>
> Our subsampling procedure for second-order information (SOI)  based on Poisson sampling implies that the expected value of the subsampled SOI is equal to the full-batch SOI at each iteration. Also, using the classical matrix concentration results, we can show that the subsampled SOI is close to the full-batch SOI with high probability as well. These two observations show that second-order information is still relevant in the minibatch version.

---

> > ### Comment · Reviewer_rkzh · 2023-08-13
> >
> > The rebuttal addressed my questions. Thank you!

---

> > > ### Author Response · Authors · 2023-08-18
> > >
> > > Thank you very much for your reply. We will revise our paper according to the constructive comments in the reviews.

---

### Official Review · Reviewer_8XFJ · 2023-07-24

**Soundness:** 3 good
**Presentation:** 3 good
**Contribution:** 3 good
**Rating:** 6
**Confidence:** 3

**Summary:**

This paper focuses on the problem of private convex optimisation using second order information where the main motivation is the acceleration of private convex optimisation problem as DP-SGD shows slow convergence. This is done in two parts. Firstly, they introduce the privatised version of the cubic method of Nesterov and Polyak when the loss function is strongly convex. In this step, the privatisation is done by solving the optimisation problem given by the global cubic upper-bound privately (DP-Solver). They analyse the algorithm and provide convergence guarantees.

Since the cubic method is computationally expensive, for the rest of the paper the authors focus on a method inspired by Newton's method with its specific application to logistic regression. Specifically, they provide a global upper bound for the logistic loss and for which the optimisation the upper bound has a closed form. They then privatise this method by privatising the steps of the optimisation algorithm. They do this in two steps. First by adding the proper amount of noise to the information from gradient and then adding noise at the update step.

Finally, they empirically compare the performance of their proposed method for logistic regression to DP-(S)GD and objective perturbation.

**Strengths:**

1. Convex optimisation appears in many settings in Machine Learning and given the increasing attention to privacy, this is a timely problem to study.

2. The combination of ideas to use a method similar to newton's method while making sure the upper bound is global like in the cubic method is nice.

3.  The convergence analysis for the proposed algorithms.

4. The details of the experiments are explained thoroughly and for the most part, they seem fair.

**Weaknesses:**

1. While private convex optimisation is an important problem and the techniques are interesting, I feel the scope of the paper is a bit limited as it only allows us to use the results for logistic regression in a private setting. For any other method that uses convex optimisation, like kernel methods, the user would need to calculate the sensitivity of the queries in algorithm 3 at which point the method being faster than DP-SGD might not be justification enough for this method.


2. One of the main motivations for this paper is the slow convergence of DP-SGD. While the authors provide references to other bottlenecks for DP-SGD like the batch size and hyper-parameter tuning, there are no references for the slow-convergence of DP-SGD and the main thing supporting this claim in the paper is Figure 1.

**Questions:**

1. The upper-bound for the convergence analysis for  ($(lw) - l(w^*)$) in both Theorem 1 and Theorem 2 is done w.r.t $w^*$ which is the optimal w not attainable in the case where $\rho>0$ as given by the lower-bound given in [BST14].  Does it not make more sense to provide the upper bound w.r.t the set of parameters $w$ which are achievable under privacy constraints?

2. Throughout the paper you mention that your proposed method is 10-40 times faster but looking at Table 1 this is the case for $\epsilon = 10$, otherwise the range is something like 3-40. Is there a reason for mentioning 10-40?

3. [KSJ18] show that the Newton's method globally converges under some conditions which hold for logistic regression. Given this result, why do we need to build the global upper bound of lemma  5.1?

4. Is the idea for building global upper  bounds for newton's something done for the first time in your paper? If not, it might be a good idea to add some references.






Minor comments:

1. You can add other examples where convex optimisation appears in ML. Some examples are kernel regression and extreme learning machines.

2. I think adding some explanation and details to Theorem 4.2 might help the reader to digest the theorem a bit better i.e. What is the value of $w$ for which the transition between the convergence rates happen? My intuition is that the privacy constraint makes a set of parameters centred around $w^*$)admissible and while $w$ is outside of this range the convergence rate is slower and once we are within those parameters the convergence is much much faster.

3. For theorem 5.6, it might help to mention that the relationship between the semi-norm and the $\ell^2$ norm for ease of understanding.




**Limitations:**

It is similar to the comment I make in the weaknesses. The authors have not clearly mentioned that while the method is faster than DP-SGD, it requires an expert to get the algorithm up and running for other convex optimisation problems.

Additionally, it is known that the computational complexity of Newton's method scales badly with dimension d which I think should be mentioned in the paper.

---

> ### Author Rebuttal · Authors · 2023-08-09
>
> We thank the reviewer for their thorough review. We respond to the main points below.
>
> > **W1:** The scope of the paper is limited to logistic regression.
>
> For simplicity and clarity, many of our results focus on logistic regression, but our techniques are more widely applicable. Logistic regression is a widely-used method and there are many DP baselines for this task, so it is an ideal case study for our methods. Indeed, our algorithm based on double noise can be applied more broadly. In Appendix C.6 (Algorithm 6), we have provided an extension of Algorithm 3 whose privacy proof holds for **every** convex, Lipschitz, and smooth loss functions. Please refer to Remark 5.5 in the main body for further discussion. Also, Algorithm 3 and its convergence analysis in Theorem 5.6 holds for every convex, Lipschitz, smooth, and doubly-differentiable GLM loss function.
> Based on your comment, we will highlight the extensions beyond logistic loss in the paper.
>
> > **W2:** Supporting claim for the slow-convergence of DP-(S)GD.
>
> Most of the theoretical literature on DP convex optimization either ignores the problem of slow convergence or makes strong assumptions (e.g., strong convexity, small diameter constrained set) to ensure rapid convergence. But it is a major issue. See [our common rebuttal](https://openreview.net/forum?id=h2lkx9SQCD&noteId=ZhStwLJeHt) for further discussion. We will emphasize these points in the next revision.
>
> > **Q1:** On the definition of excess loss with privacy constraints.
>
> Our results follow the standard formulation in terms of excess loss used in the literature. It would be interesting to compare our upper bound with the optimal excess loss attainable under DP, but this would be difficult as the latter quantity is not known exactly. E.g., the lower bounds of BST14 are asymptotic and do not give tight leading constants. However, the excess error of our cubic Newton’s method is (near) optimal: the lower bound for the class of strongly convex functions [BST14,Thm.5.5] implies that the achievable excess error has the optimal dependence on the dimension, privacy budget, and number of samples up to a log factor.
>
> > **Q2:**  10-40 times faster / 3-40 times faster
>
> Thanks for noticing this oversight. We neglected to update this number after adding further experiments. We have updated the abstract and mentioned that our method is 3-40 times faster in general.
> Our method shows the greatest improvement for datasets such as Adult or Covertype where the logistic loss has an ill-conditioned Hessian.  For well-conditioned synthetic data, there is less room for improvement.
>
> > **Q3:** On the comparison with the results of [KSJ18].
>
> The results of [KSJ18] show a global convergence for the “damped” Newton method, i.e., Newton's method with **non-unit** step size. However, it is not clear how the algorithm of [KSJ18] can be used for logistic regression in an unconstrained setting. By the results in Section 2.3 Part (a) in [KSJ18], the step size is proportional to $\exp(-\|\|\text{optimal solution}\|\|)$. For unconstrained logistic regression, there is no a priori knowledge on the norm of the optimal solution, therefore it is not clear how this result can be used. We have included a detailed comparison with [KSJ18] in the revised version of the paper.
>
> > **Q4:** Novelty of the quadratic upper bound.
>
> To the best of our knowledge our quadratic upper bound for the logistic loss is a novel optimization technique. We will emphasize this point.
>
>
> > **minor comments**
>
> Thanks! We have included more examples in the introduction.
>
> The transition point in Theorem 4.2 happens when  $\|\|w_t - w^\star\|\|$ is less than $ \frac{3 \mu }{4 L_2} $ where $\mu$ and $L_2$ are the strong convexity parameter and Hessian Lipschitzness constant, respectively. Your intuition is correct and we will provide more discussion on this point after the statement of Theorem 4.2.
>
> Relationship between semi-norm and $\ell_2$ norm:
> V is the projection matrix on the subspace spanned by the training set. Therefore, for every point $x \in \mathbb{R}^d$, its semi-norm, i.e., $\|\|x\|\|_V \leq \|\|x\|\|_2$. The main reason behind proving the convergence results in $\|\|x\|\|_V$ is that the components of the output vector, i.e., $w_T$, outside the subspace spanned by the data do not affect the excess error.

---

> > ### Comment · Reviewer_8XFJ · 2023-08-14
> >
> > The rebuttal has addressed my questions and comments. Thanks.

---

> > > ### Author Response · Authors · 2023-08-18
> > >
> > > Thanks a lot for your response! We'll revise our paper based on the constructive feedback from the reviews.

---

### Author Rebuttal · Authors · 2023-08-09

We are extremely grateful to the reviewers & AC for their time & comments. Their comments have been very constructive. We respond to each review individually, but we respond to some common points here:

**On the slow convergence of DP-SGD:** Several reviewers questioned the claim that DP-(S)GD converges slowly, which we made in the last paragraph on the first page of the submission and supported with Figure 1. We wish to elaborate on this point, since it is integral to the motivation for our work. Informally, this is because the addition of noise can push us in directions of increasing loss, so we need more conservative step sizes to avoid moving too far in the wrong direction. This fact is reflected both theoretically and empirically in the literature:

 - Theoretically, optimal instantiations of DP-SGD use a smaller step size than in the non-private case, e.g. [Bassily et al. (2019)](https://arxiv.org/pdf/1908.09970.pdf) use $\eta$ that decays as $\max\\{1/\sqrt{n}, \sqrt{d}/\varepsilon n\\}$, much smaller than the non-private setting of $\eta = 1 / \beta$ when the loss is $\beta$-smooth. Smaller step size requires more steps (i.e. more iterations) to converge. Also, refer to the attached PDF (Item 4) for an example of noiseless and noisy gradient steps on a 1-smooth quadratic loss. This example intuitively explains why we need small $\eta$ for DP-(S)GD.

 - Empirically, e.g., Figure 1 of [Kurakin et al. (2022)](https://arxiv.org/abs/2201.12328) shows that the accuracy of DP training significantly improves if we run DP-SGD for more iterations while keeping the overall privacy budget fixed.

**On computing second-order information and iteration/runtime complexity:** Several reviewers had concerns relating to the complexity of Algorithm 1. We focused on oracle complexity, which is common in the literature. We can also show that in terms of computational complexity, Algorithm 1 is competitive with the best first-order methods. We will add a discussion of this to the paper.

First, we do not need to compute a full Hessian to run DPSolver, but only to compute Hessian-vector products (HVPs). HVPs in practice can often be computed in the same order of time as it takes to do a gradient computation (see e.g. https://jax.readthedocs.io/en/latest/notebooks/autodiff_cookbook.html). So for any first-order method we use as DPSolver, each iteration of Algorithm 1 can be made to take time comparable to the runtime of DPSolver on an arbitrary loss.

While we use DP-GD as DPSolver for simplicity of presentation, any first-order method with the same privacy and utility guarantee as DP-GD gives the same privacy and utility guarantee for Algorithm 1, so we can choose e.g. a more efficient method for strongly convex losses as Algorithm 1. As a concrete example, we can use Alg.1 in [Wang et al. (2017)](https://proceedings.neurips.cc/paper/2017/file/f337d999d9ad116a7b4f3d409fcc6480-Paper.pdf) using which the subproblem solver has HVP and gradient complexities of order $n\log(n)$ and iteration complexity of order $\log(n)$.


Putting it all together, the runtime of Algorithm 1 in practice will be no worse than $O(\log\log n)$ times the runtime of the first-order method we choose as DPSolver. Furthermore, we hypothesize that since we are using DPSolver to optimize a well-behaved cubic function, the runtime of DPSolver can be made even faster, making our Algorithm 1 even faster as well.

---

> ### Comment · Reviewer_Xt6K · 2023-08-12
>
> > Putting it all together, the runtime of Algorithm 1 in practice will be no worse than $\mathcal{O}(loglog(n))$ times the runtime of the first-order method we choose as DPSolver.
>
> Does this mean that theoretically, the runtime of the proposed method is indeed worse than the DP-GD or other first order methods?

---

> > ### Author Response · Authors · 2023-08-12
> > **Reply to Reviewer Xt6K regarding the Runtime of Our Algorithm 1**
> >
> > The standard way to theoretically analyze runtime in the optimization theory literature is to consider *oracle complexity*. **Our Algorithm 1 has lower oracle complexity than is possible with first order methods.** That is the point of the result; we did not try to optimize any other complexity measure.
> > If one is interested in the total runtime, then the experimental results are more informative.
> >
> > The oracle complexity of Algorithm 1 does not depend on the DPSolver subroutine. However, the total runtime is $O(\log\log n)$ times the cost of computing a gradient & Hessian plus running DPSolver. For simplicity, we used a first order method as the DPSolver subroutine; it is possible that there are faster subroutines we could use instead. (The cubic function is a much simpler function than the worst-case scenario that witnesses the lower bounds, so it seems likely we could improve.) But we did not explore this, since it doesn't improve the oracle complexity.
> >
> > Comparing the total runtime of Algorithm 1 to first order methods will depend on many factors.
> > Suppose the dataset size $n$ is large compared to other parameters like the dimension $d$. (I.e. $n \to \infty$ and $d = n^{o(1)}$.) Then the most expensive operation in Algorithm 1 is to compute gradients and Hessians. Note that the runtime of DPSolver is independent of $n$. In this case, our algorithm will outperform first order methods.
> >
> > On the other hand, if the dimension $d$ is relatively large compared the dataset size $n$, then it gets messy: The runtime of the DPSolver subroutine may dominate and, if the DPSolver subroutine is itself a first order method, then it seems likely that it would be faster to directly apply a first order method. (In general, second order methods do not work well when the dimension is large.)

---

> ### Comment · Reviewer_FNjC · 2023-08-14
>
> > much smaller than the non-private setting of $\eta=1/\beta$ when the loss is $\beta$-smooth.
>
> I don't think the stepsize is $1/\beta$ for SGD in the non-private setting, which is not enough to control the variance of the stochastic gradient. It should scale with $1/\sqrt{T}$ for smooth convex/nonconvex losses. Do you mean GD?

---

> > ### Author Response · Authors · 2023-08-14
> > **Reply to Reviewer FNjC**
> >
> > Thanks for catching that; indeed this step size is for full-batch GD.
> > In trying to condense our reply we accidentally compared a private, stochastic step size with a non-private, non-stochastic step size. However, for both the full-batch and stochastic settings, it is true that the addition of Gaussian noise for privacy requires a smaller step size to guarantee that the loss decreases in expectation when compared to non-private GD, as Figure 4 demonstrates.

---

### Decision · Program_Chairs · 2023-09-21

**Decision:**

Accept (poster)

**Comment:**

After a fairly intense discussion, all reviewers recommended some degree of acceptance of the paper. However, I **strongly** encourage the authors to update their manuscript to reflect the issues raised by the reviewers and the updates presented in their rebuttal.

Specifically, during the AC-reviewer discussion phase, the reviewers appreciated the contributions the paper makes to the field of differential privacy but still highlighted some areas for improvement. One concern is the lack of a clear explanation about the limitations of the methods used, e.g., the scalability of Newton's method. Reviewers also suggested that a **brief** review of related optimization literature would be welcome, particularly to further motivate the paper (as opposed to only Fig. 1). Finally, reviewers suggested streamlining and improving the statement of the theorems.